# Factors that affect migratory Western Atlantic red knots (*Calidris canutus rufa*) and their prey during spring staging on Virginia's barrier islands

**Erin L. Heller**[1¤]*, **Sarah M. Karpanty**[1]*, **Jonathan B. Cohen**[2], **Daniel H. Catlin**[1], **Shannon J. Ritter**[1], **Barry R. Truitt**[3], **James D. Fraser**[1]*

**1** Department of Fish and Wildlife Conservation, Virginia Tech, Blacksburg, Virginia, United States of America, **2** Department of Environmental and Forest Biology, The State University of New York, Syracuse, NY, United States of America, **3** The Nature Conservancy, Virginia Coast Reserve, Nassawadox, VA, United States of America

¤ Current address: Biology Department, Randolph College, Lynchburg, VA, United States of America
* elheller@vt.edu (EH); karpanty@vt.edu (SK); fraser@vt.edu (JF)

**Data Availability Statement:** Data available from the Virginia Coast Reserve's Environmental Data

## Abstract

Understanding factors that influence a species' distribution and abundance across the annual cycle is required for range-wide conservation. Thousands of imperiled red knots (*Calidris cantus rufa*) stop on Virginia's barrier islands each year to replenish fat during spring migration. We investigated the variation in red knot presence and flock size, the effects of prey on this variation, and factors influencing prey abundance on Virginia's barrier islands. We counted red knots and collected potential prey samples at randomly selected sites from 2007–2018 during a two-week period during early and peak migration. Core samples contained crustaceans (Orders Amphipoda and Calanoida), blue mussels (*Mytilus edulis*), coquina clams (*Donax variabilis*), and miscellaneous prey (horseshoe crab eggs (*Limulus polyphemus*), angel wing clams (*Cyrtopleura costata*), and other organisms (e.g., insect larvae, snails, worms)). Estimated red knot peak counts in Virginia during 21–27 May were highest in 2012 (11,959) and lowest in 2014 (2,857; 12-year peak migration $\bar{x}$ = 7,175, SD = 2,869). Red knot and prey numbers varied across sampling periods and substrates (i.e., peat and sand). Red knots generally used sites with more prey. Miscellaneous prey ($\bar{x}$ = 2401.00/m², SE = 169.16) influenced red knot presence at a site early in migration, when we only sampled on peat banks. Coquina clams ($\bar{x}$ = 1383.54/m², SE = 125.32) and blue mussels ($\bar{x}$ = 777.91/m², SE = 259.31) affected red knot presence at a site during peak migration, when we sampled both substrates. Few relationships between prey and red knot flock size existed, suggesting that other unmeasured factors determined red knot numbers at occupied sites. Tide and mean daily water temperature affected prey abundance. Maximizing the diversity, availability, and abundance of prey for red knots on barrier islands requires management that encourages the presence of both sand and peat bank intertidal habitats.

Initiative (https://doi.org/10.6073/pasta/
9d6961262dfdaee98720f118f5ad816c, Karpanty
et al. 2021).

**Funding:** This research was supported by the
National Marine Fisheries Service of the National
Oceanic and Atmospheric Administration within the
United States Department of Commerce (https://
www.fisheries.noaa.gov/ SMK, JDF), the National
Science Foundation (NSF) Virginia Coast Reserve
Long Term Ecological Research Grants DEB-
1237733 and DEB-1832221 (https://www.nsf.gov/
SMK), Virginia Department of Wildlife Resources
(https://dwr.virginia.gov/ SMK, JBC, JDF), the U.S.
Fish and Wildlife Service (https://www.fws.gov/
ecological-services/ SMK, JBC, DHC, JDF), Sigma
Xi Grants-In-Aid of Research (https://www.sigmaxi.
org/programs/grants-in-aid ELH), the Virginia Tech
Graduate Resource Development Program (https://
gsa.vt.edu/programs/grdp.html ELH), and the NSF
Graduate Research Fellowship grant DGE-1651272
(https://www.nsf.gov/ ELH). Any opinion, findings,
and conclusions or recommendations expressed in
this material are those of the authors and do not
necessarily reflect the views of the National Science
Foundation, or of other sponsors. The funders had
no role in study design, data collection and
analysis, decision to publish, or preparation of the
manuscript.

**Competing interests:** The authors have declared
that no competing interests exist.

## Introduction

Species conservation requires an understanding of the factors that influence a species' distribution and abundance [1,2]. Understanding the factors that affect migratory species throughout their annual cycles is challenging, though of great importance [3,4]. Staging sites enable animals to migrate long distances in large jumps over relatively short time periods by providing abundant and predictable food so migrants can replenish fat stores and rest mid-migration [4–9]. Migratory shorebirds must be adaptable to variation in food availability on staging sites [10,11]. Anthropogenic and climate–related factors have led to the rapid loss and degradation of staging sites, amplifying the importance of remaining habitat and food availability for migratory shorebirds [12–18].

The Western Atlantic red knot (*Calidris canutus rufa*; 'red knot'), listed as federally-threatened in the United States (U. S.) and endangered in Canada, has one of the longest migrations in the western hemisphere, travelling from boreal wintering grounds as far south as Tierra del Fuego, Argentina, to breeding grounds in the Canadian Arctic [19,20]. Due to its imperiled status, recognized after large declines in the mid-1990s [21–24], and long migration, red knot staging ecology often is viewed as an exemplar of the challenges faced by long–distance migrant shorebirds [25,26].

Large numbers of red knots historically frequented North American stopover sites on coastal beaches from Florida to Massachusetts [12,27]. However, while thousands of red knots continue to use coastal beaches on the United States' Atlantic Coast [23], most of them now stop at two staging sites each spring migration: the Delaware Bay and Virginia's barrier islands [20,28,29]. Despite the relatively short distance between the two sites (~125 km), only about 5% of red knots move between these sites in a given year, and red knots remain at both sites for approximately 2 weeks [29]. Additionally, Cohen et al. [29] found that radiotagged red knots rarely left the site where they were tagged before late May to early June, the time when most red knots on the Atlantic Coast leave for the Arctic. These findings suggest that Virginia and the Delaware Bay are primarily independent staging sites within years.

Historically, much attention was given to red knots using the Delaware Bay spring staging site [29–31], as it consistently supports between about 50–70% of the annual spring migrating population of red knots, at higher densities compared to Virginia's barrier islands (on average 291 red knots/km shoreline in Delaware Bay vs. 81 red knots/km in Virginia [29–31]). The large number of red knots using the Delaware Bay staging site each spring usually is attributed to the abundance of Atlantic horseshoe crab eggs (*Limulus polyphemus* [30–37]).

Although red knots feed primarily on horseshoe crab eggs in the Delaware Bay [30,32–34], they feed on hard-shelled bivalves throughout most of their migration and boreal wintering ranges [38–40]. The red knot's reliance on bivalves includes its Virginia spring staging site, where coquina clams (*Donax variabilis*) and blue mussels (*Mytilus edulis*) have been described as the most abundant and used prey resources by red knots [31,41]. Further, Heller [42] used fecal DNA metabarcoding analyses to confirm that red knots consumed bivalves (Orders Venerida and Mytiloida) in Virginia and found that red knots also consumed crustaceans (Orders Amphipoda and Calanoida) and insect larvae (Order Diptera).

Numerous studies have demonstrated positive correlations between waterbird numbers and invertebrate prey [14,28,43–45]. Because red knots likely track prey resources in space and time [32,46], and different prey species may be available in different habitats at different tides, red knots may shift foraging locations throughout the day to maximize foraging efficiency [47]. Some uncertainty remains concerning the influence that prey may have on red knot distribution and flock size on Virginia's barrier islands. For example, Cohen et al. [41] suggested that coquina clams, which live primarily in sand, were the dominant prey item for red knots in

Virginia. Watts and Truitt [26], however, suggested that by the start of the red knot's peak migration, red knots used peat banks, where blue mussels and other prey primarily live, ten-times more than sand. Although studies have been conducted on the prey that red knots potentially consume in Virginia [26,41], these were limited to short–term datasets ($\leq$ 2 years) and did not fully address all potential prey resources.

To gain a better understanding of how many red knots use Virginia's barrier islands each spring migration and how numerous factors affect both red knot and prey abundance, our objectives were to determine: 1) the peak counts annually of red knots using Virginia's barrier islands during peak migration (May 21–27) from 2007–2018, 2) if prey other than coquina clams and blue mussels were available to red knots over an extended (> 2 years) time, 3) if and how red knot and prey numbers varied between sampling periods and substrates, 4) if there were differences in prey abundances between sites used and unused by red knots, 5) if red knot presence and flock size were influenced by prey in Virginia, 6) if red knot presence and flock size in Virginia were influenced by an index of the number of red knots in the flyway (i.e., using boreal wintering red knot counts in Tierra del Fuego as one index for the flyway population), and 7) what factors affected prey abundance in Virginia over time and space.

## Study area

We studied red knots on eleven barrier islands in the Virginia Coast Reserve Long-Term Ecological Research site from Assawoman Island in the north to Fisherman Island in the south (Fig 1; 37°23.7'N, 75°42.5'W [29]). The combined barrier island shoreline extends approximately 82 km and is bounded by the Atlantic Ocean on the east and a shallow lagoon system with open water, mudflats, and *Spartina spp*. marsh on the open bays to the west, between the barrier islands and the Delmarva Peninsula mainland [29,48]. The islands are separated from each other by a series of channels and shallow marshes that follow Virginia's coastline within the Delmarva Peninsula [49]. The islands are predominantly uninhabited and undeveloped, accessible primarily by boat, and experience little anthropogenic activity.

We sampled red knots and invertebrate prey on 11 of these barrier islands, which together comprise approximately 82 km of Atlantic Ocean intertidal shoreline, from Assawoman Island in the north to Fisherman Island in the south, May 2007–2018. Basemap content is the intellectual property of Esri and is used herein with permission (Copyright © 2022 Esri and its licensors. All rights reserved). The inset aerial image from 2021 was acquired from the U.S. Department of Agriculture's National Agriculture Imagery Program (NAIP), available at [50] from the Aerial Photography Field Office and shows an example of the location of 5 sample locations on peat banks sampled during early migration on Metompkin Island, 2018 [51]. The locations of all sample points from 2018 are visualized in S1 Fig. The geographic coordinates and details on all sample points from 2007–2018 can be found [51].

The islands provide foraging habitat for migratory shorebirds preparing to breed or to continue migrating to breeding grounds farther north. These shorebirds forage principally on two ocean intertidal substrates: sand and peat [26,41,52,53]. Large populations of coquina clams are found in sandy intertidal zones, the most common intertidal substrate on Virginia's barrier islands [41,54]. Intertidal peat banks, which comprise about 6% of the shoreline, are formed when outer beaches erode along low and narrow island segments that transgress over tidal marshes on the bay side of the islands [55,56]. Although these peat banks are found sporadically across only some islands, they may support dense blue mussel and other invertebrate populations [26,41,55]. The islands vary in their susceptibility to overwash and landward-movement based partly on morphological characteristics, such as elevation. Low elevation islands, in general, are narrow and lack vegetated dunes. These islands are thus prone to

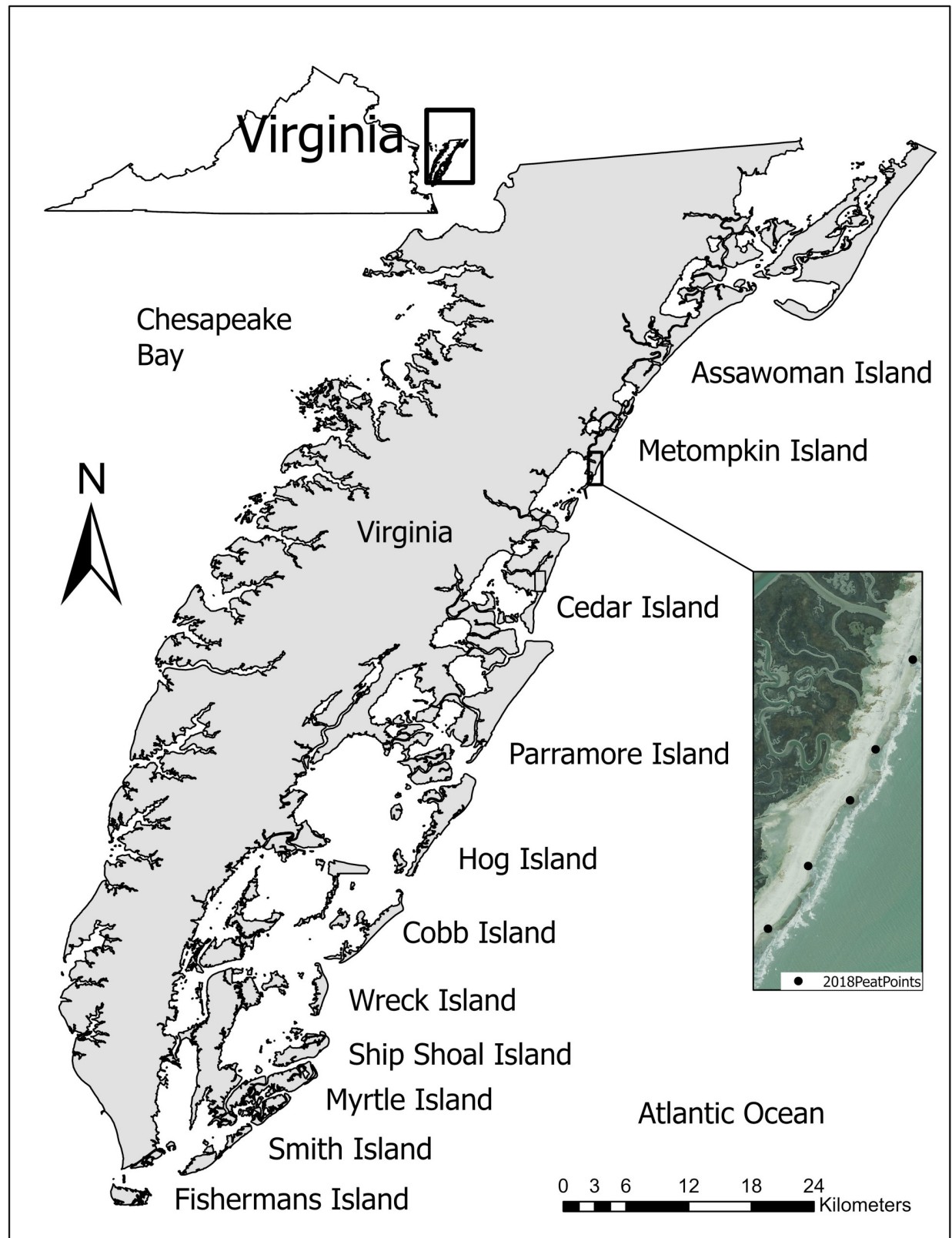

**Fig 1. Study area on the Virginia Coast Reserve Long Term Ecological Research site consisting of barrier islands off the Eastern Shore of Virginia.**

overwash, enabling the formation of peat banks on the islands' ocean side over time. Higher elevation islands have well-developed, more vegetated dunes that help resist overwash events and thus generally lack extensive peat banks [57,58]; however, high elevation islands can sustain peat banks on lower elevation, narrow island segments.

## Methods

### Early and peak migration sampling of red knots and prey

We collected data on red knots and their invertebrate prey along the waterline of the Atlantic Ocean intertidal zones on Virginia's barrier islands each year from 2007–2018 (Figs 1 and S1, [51]). Data collection occurred over two time periods from May 14–20 ('early migration period') and May 21–27 ('peak migration period') 2007–2018, corresponding to the two weeks of the red knot staging period during which red knots most abundantly use Virginia's barrier islands [29]. Cohen et al. [29] detailed that red knots use the Virginia barrier islands as staging habitat from late April to early June each year, with numbers building through early May and peaking during the week of May 21–27. Prior studies of radio-tagged red knots in Virginia [29] also detailed that red knots used both peat and sandy substrates for foraging, but that peat substrate was only available to the foraging red knots within 2-hrs of low tide.

Due to the logistical challenges of accessing peat substrate within two hours of low tide on these remote island locations, we had to dedicate approximately one week of field effort each year to sample this substrate type only to obtain adequate sample sizes for analyses of red knot and prey abundances on peat substrate [41]. Given the remote locations of these islands, each boat and field sampling crew could sample peat substrate only on one island per day given the time to boat from the mainland to the island, and then to walk from a safe landing point to the location of the peat substrate sampling areas, and then to walk back to the boat, all within 2 hours of low tide and within the constraints of sunrise and sunset times. We began sampling approximately 50 random points per year along the waterline on peat substrate exclusively during the red knot's early migration (May 14–20; 'early migration period' based on [29]) in 2008. We sampled the random points on this peat substrate from two hours before low tide to two hours after low tide, encompassing the last hour of falling tide, low tide, and the first hour of rising tide, when peat was most exposed. Then, during the peak period of red knot migration in Virginia (21–27 May [29]), we sampled a new set of approximately 100 random points along the waterline each year that fell in either substrate, sand or peat, depending on the tide state at the time the sampling crews arrived at a point. Sampling by the field crews during the peak migration sample period (21–27 May) was conducted irrespective of tide state; specifically, the crews left the mainland in boats near sunrise each day irrespective of tides. Given that only 6% of the Virginia shoreline is peat substrate, most samples during peak migration fell on sandy substrate, with fewer on peat substrate during peak migration.

The 50 random sample points on peat substrate during early migration each 14–20 May and the 100 random points on sand or peat substrate during peak migration each 21–27 May were generated to fall at least 200 m away from each other random point along the ocean intertidal zone of each island each year. We used the Hawth's Tools extension [59] in ArcGIS 10.1 [60], the Geospatial Modelling Environment extension [61] in ArcGIS 10.5 [62], and the most recently available United States Department of Agriculture (USDA) Farm Service Agency's National Agriculture Imagery Program (NAIP) orthophotography imagery [63] to generate these random sample points. The locations of randomly generated points varied by year and the geographic locations for each point for each year are available in [51] and an example of points from 2018 is available in S1 Fig. To determine the location of peat substrate on which to generate random points each year for the early migration sampling period, we walked along

the shore at low tide when peat substrate was exposed and recorded the location of the north and south boundaries of peat banks greater than 1 m in length using hand-held GPS devices. These data were cross-referenced with orthophotography imagery and were jointly used to stratify random sampling points by substrate type. Not all islands contained peat banks in any given year, and the number of points per island varied by island length.

Field sampling crews would navigate to each random point and count the number of non-flying red knots, if present, within a 100 m radius semicircle of each point placed on the water line. After red knots were counted, if present, or immediately if no red knots were present, we sampled invertebrate prey availability by collecting a core sample of the substrate at the water-line at each sampling point using a section of PVC piping (10 cm diameter x 3.5 cm deep; core volume = 275 cm$^3$). Thus, invertebrate prey samples were collected at all points, those with red knots present and those without red knots present. We report prey results as number of organisms/m$^2$ to be comparable to other red knot prey studies and thus assume for purpose of calculation that all prey are on the surface so that the area sampled in each core is 0.00785 m$^2$. The core's depth represented the approximate length of a red knot's bill based on museum specimens [64], so that we sampled only prey that red knots would be able to access while foraging. The PVC was pushed into the substrate until the top was even with the substrate's surface, then a trowel was slipped under the PVC to prevent the substrate sample from falling out as the PVC and core sample were retrieved. Samples were placed in a zipper-lock plastic bag, returned to the lab, and frozen for future identification. We used a series of sieves, with the smallest mesh size #40 (0.32 mm holes), and a dissecting microscope to sort and count the number of invertebrates in each sample. We sorted prey by category (i.e., crustacean (Orders Amphipoda and Calanoida [41]), blue mussel, coquina clam, miscellaneous–horseshoe crab eggs, angel wing clams (*Cyrtopleura costata*), insect larvae, snails, worms that we were unable to identify to species). We grouped these organisms as "miscellaneous" due to the relatively low number of animals of each type collected.

### Sand and peat tidal sampling of red knots and prey

To address how red knots and prey varied at the same location over the tidal cycle, we sampled red knots and prey at the water line as the tide advanced and receded on sand (2013–2018) and peat banks (2015–2018). Sampling dates occurred between May 17–28. For sand tidal sampling, we counted the number of non-flying red knots, if present, within a 100 m radius semicircle on the water line every hour for 12 hours, coinciding to the full tidal cycle. After counting red knots, if present, or immediately if not present, we collected a core sample at the water line. Each core sample was collected at a point directly in line with the previous point, perpendicularly from the water's edge, accounting for the changing water line as the tide receded or advanced. The same methods were used for peat tidal sampling within the four-hour period during which peat banks were most exposed: two hours before low tide to two hours after low tide.

### Prey spatial tidal sampling on peat and sand

To determine spatial variation of prey across the tidal cycle, we conducted spatial tidal sampling on peat banks (May 17, 2019; Myrtle Island) and sand (May 20, 2019; Hog Island). We collected core samples from a central location (0 m) and 5 m and 10 m in North and South from the central location, along the water line, every hour. Peat spatial tidal sampling occurred from two hours before to two hours after low tide, the period when peat was exposed. Due to time constraints, sand spatial tidal sampling was conducted for only 5 hours between the rising and falling tides, as opposed to during the full 12-hour tidal cycle.

### Ethics statement

All research was conducted in compliance with the laws of the United States of America and followed the protocols of the Eastern Shore of Virginia National Wildlife Refuge (ESVNWR) General Activities Special Use Permit # G19-04, ESVNWR Fisherman Island National Wildlife Refuge Research and Monitoring Special Use Permit # R19-0, Virginia Department of Game and Inland Fisheries Scientific Collection Permit # 064944, Chincoteague National Wildlife Refuge Research and Monitoring Special Use Permit # 2019–010, Virginia Department of Conservation and Recreation Division of Natural Heritage Natural Area Preserve Research and Collecting Permit # RCP-ESR01-19 (renewal for ESR03-18), The Nature Conservancy Research Permit, Commonwealth of Virginia Marine Resources Commission Permit # 19–040, and Virginia Tech Institutional Animal Care and Use Committee permit number 16–244 (FWC). The islands sampled are protected and monitored by The Nature Conservancy, the United States Fish and Wildlife Service, the Virginia Department of Game and Inland Fisheries, and the Commonwealth of Virginia's Department of Conservation and Recreation's Natural Heritage Program.

### Data analyses

**Red knots over time and space.**   Shapiro-Wilk normality tests indicated that red knot (W = 0.43, p < 0.001) and all prey abundance (W = 0.53, p < 0.001) data were not normally-distributed during the early migration sample period (14–20 May). Shapiro-Wilk normality tests also indicted that red knot (W = 0.23, p < 0.001) and all prey abundance (W = 0.37, p < 0.001) data were not normally-distributed during the peak migration sampling period (21–27 May). Therefore, we used Wilcoxon rank sum tests with Bonferroni correction to determine if there were differences in red knot and prey numbers (i.e., crustacean, blue mussel, coquina clam, miscellaneous prey, all prey) between 1) early and peak migration periods and 2) sites used vs. unused by red knots. We used a Pearson's chi-square test to determine if mean available prey from core samples differed in their community make-up on peat vs. sand during the peak migration period. Then we used Wilcoxon rank sum tests with Bonferroni correction to determine if any differences existed in mean prey abundances on peat and sand substrates within the peak migration period.

**Early and peak migration periods–red knot models.**   We used zero-inflated negative binomial mixed-effects regression models to determine factors affecting red knot presence and flock size in Virginia, as our data contained more 0s than expected based on the negative binomial distribution and were overdispersed during both early and peak migration periods ($\bar{x}$ red knot flock size range: 0.14–29, variance range: 0.17–16,247, [65,66], S1 and S2 Tables). These models included two processes. The zero-inflated process addressed the likelihood of observing more 0s (red knot absences) than expected given the covariates under the assumed negative binomial distribution. The count process measured the flock size of red knots (i.e., the number of red knots at a given sampling point), which can include 0, conditional on the zero-inflated part of the model [67]. We analyzed the effects of covariates on the likelihood of 1) more site absences by red knots than expected based on the negative binomial distribution [68] and 2) the abundance of red knots per 100 m radius semicircle on the water line at the sampling point ('flock size') during both early (2008–2018, except 2010) and peak migration (2007–2018) periods.

Explanatory variables considered included prey abundance (number/m²), tide, distance to roost (m), and counts of red knots at the Tierra del Fuego boreal wintering grounds within the same year. The counts of red knots at the Tierra del Fuego boreal wintering grounds served as an index for the total number of red knots in the flyway [21,22]. Red knots marked in locations

across the flyway have been resighted on Virginia's barrier islands [69], but long-term and robust counts of red knots in other locations outside of Delaware Bay were unavailable for comparison across the time span of this study. Thus, the counts of Tierra del Fuego birds were used as an index of red knots in the flyway since the Tierra del Fuego count data fully overlaps the time period of this study and was independent from the U.S. mid-Atlantic Coast region in which both Delaware Bay and Virginia staging sites exist. We characterized tide (i.e., high, falling, low, rising) from each sampling observation as: high within 1 hour of the predicted high, falling from 1 hour after the predicted high to 1 hour before the predicted low, low within 1 hour of the predicted low, and rising from 1 hour after the predicted low to 1 hour before the predicted high [41]. High tide was not sampled during the early migration period, as peat banks typically were not exposed at high tide. Distance to roost was calculated as the distance between each sampling point and known red knot night roosts on Chimney Pole and Wreck Island [29,41]. The closer of the two distances/sample point was used for our analyses. We compared all continuous covariates using Pearson correlation coefficients to identify highly correlated covariate combinations (represented by values > |0.7| [70,71]). Highly correlated combinations for early migration period and peak migration period were not included in our model subsets (S3 Table). For all models, we included island and year as random effects to prevent pseudoreplication of intra-island sampling units over time [72].

We used an information-theoretic approach (i.e., Akaike information criterion [73]) by building an *a priori* candidate model set of sixty-eight models. We ranked these zero-inflated negative binomial mixed-effects models using Akaike's Information Criterion for small sample sizes (AIC$_c$), with lower AIC$_c$ values indicating better-supported models. Here, we report all models with ΔAIC$_c$ < 4 [74]. Full model sets are in S4 Table. Three models did not converge during early migration period and were thus removed (S4 Table). We calculated goodness-of-fit values based on Nagelkerke [75].

**Early and peak migration periods–prey models.** We used generalized linear mixed-effects regression models to determine factors affecting potential red knot prey resources. We analyzed the effects of covariates on different prey abundances per core sample during both early (2008–2018, except 2010) and peak migration (2007–2018) periods. Explanatory variables included tide, mean daily water temperature (˚C), substrate (i.e., sand or peat), and island type (i.e., high or low elevation). We characterized tide (i.e., high, falling, low, rising) from each sampling observation as described above for red knot models. Mean daily water temperatures were collected from buoy 44009 (Delaware Bay, DE; 38.457˚N, 74.702˚W), maintained by National Oceanic and Atmospheric Administration's (NOAA) National Data Buoy Center (NDBC [76]). This buoy is located south of the Delaware-Maryland boundary (46 km southeast of Cape May, New Jersey) and was selected as it was the closest buoy to our sampling area that had ocean temperature data representing the entirety of our sampling years (2007–2018). The buoy was 98.01 km north of Assawoman Island and 188.29 km north of Fisherman Island. Substrate was not included in early migration period modeling, as only peat substrate was sampled. Island type (i.e., high vs. low) was based off classifications done in Wolner et al. [57] and Vinent and Moore [58] and our own visual assessments: Assawoman Island (high), Metompkin Island (low), Cedar Island (low), Parramore Island (high), Hog Island (high), Cobb Island (low), Wreck Island (low), Ship Shoal Island (low), Myrtle Island (low), Smith Island (high), and Fisherman Island (high). For all models, we included island and year as independent random effects to prevent pseudoreplication of intra-island sampling units over time.

We used an information-theoretic approach (i.e., Akaike information criterion) by building an *a priori* candidate model set of 8 models for early migration period and 16 models for peak migration period. We ranked these models using Akaike's Information Criterion for small sample sizes (AIC$_c$), with lower AIC$_c$ values indicating better-supported models. Here, we

report all models with $\Delta AIC_c < 4$. Full model sets by prey type and sampling period are in S5 Table. We calculated goodness-of-fit values based on Nakagawa and Schielzeth [77].

**Red knot peak count numbers.** To estimate a peak count of red knots using the entire length of Virginia's barrier islands during peak migration (May 21–27) each year, based on our observations at random sampling points across the island chain (Raw Data Expansion), we used the following equation to linearly expand our predictions from our sampling points:

$$\text{Peak Count Red Knots Per Year}_{2007-2018} = \frac{\text{Total Shoreline Length (m)}}{\sum(n1 + n2 + \ldots n11) * 200 \text{ m}} * (\sum k_1 + k_2 \ldots k_{11}),$$

where, $n_{1-11}$ = number of sampling points on each of the 11 islands, and

$k_{1-11}$ = sum of all red knots counted in all sampling points on each of the 11 islands.

Additionally, to account for non-normality and variation in island use, we used zero-inflated negative binomial regression models (ZINB Model Expansion; Red Knot ~ Island + Year) using Program R package pscl to predict the number of red knots per sampling point for each island-year combination. We used this type of model because our data contained more 0s than expected based on the negative binomial distribution and were overdispersed ($\bar{x}$ red knot flock size range for peak migration: 9–29, variance range: 778–16,247, S1 and S2 Tables). To estimate standard errors for the model predictions, we first used the vector of the regression parameter estimates and their variance-covariance matrix in a Monte Carlo simulation with 1,000 iterations to separately calculate the mean and variance of the count and zero inflation parts of the peak count values for each island/year combination on the linear scale. We then used the delta method to calculate the standard error of each peak count value on the count scale, as well as the sum of these peak counts across islands within each year.

For both methods, we estimated the length of each island and the total shoreline using aerial imagery from USDA's Farm Service Agency's NAIP orthophotography imagery [63] and our own GPS estimates from walking the islands during the study. We compared the predictions from both methods (Raw Data Expansion and ZINB Model Expansion) using a paired t-test over all years, as the predictions for both methods were normally distributed (Shapiro-Wilk; W = 0.95, p = 0.57 and W = 0.95, p = 0.68, respectively). We used a Freidman test to determine if there were differences in red knot numbers by year (2007–2018), controlling for island. We used Pearson correlation to determine the strength and direction of any potential trends in the peak count number of red knots in the Virginia over time (2007–2018) and of any relationship between peak count red knot numbers in Virginia and the peak count numbers in the Delaware Bay staging sites (Shapiro-Wilk; W = 0.92, p = 0.31). We used aerial flight or ground-based peak count numbers of red knots in Delaware Bay for these comparisons with Virginia red knot peak count numbers as the Delaware Bay peak count numbers were available for the entire time period analyzed here (2007–2018), whereas more recent mark-resight based estimates of red knot total staging population numbers are only available since 2011 [78–80].

The resulting counts of red knots in Virginia during peak migration using both methods make three assumptions and should not be interpreted as total staging population estimates [29,78] during peak migration until these assumptions are further tested. First, we assumed that the number of birds entering and exiting Virginia's barrier islands were equal during peak migration. While we would need resighting data to formally test this assumption, red knot behavior demonstrates that the majority of red knots arrive by the start (May 21) of the peak migration period (May 21–27) and that red knots typically remain on the islands for at least two weeks, reducing the likelihood of immigration and emigration during this time [25,28,61]. Second, we assumed that red knot detectability and identification were perfect over the 12-years of this study. While ZINB models do not account for imperfect detection or

identification [81], our field methodology was conducted using only observers trained extensively in shorebird counting and identification, reducing the likelihood of missing or incorrectly identifying red knots. Third, our predictions assumed that no birds were double counted within a given year's peak migration. We minimized the potential bias of this assumption by moving linearly down the beach's oceanfront when counting red knots and staying at least 100 meters away from flocks to prevent dispersal.

**Peat and sand tidal sampling of red knots and prey.** Shapiro-Wilk normality tests indicated that red knot and all prey abundance data during tidal sampling were not normally-distributed on peat (red knot: W = 0.60, p < 0.001; all prey: W = 0.75, p < 0.001) and sand (red knot: W = 0.60, p < 0.001; all prey: W = 0.79, p < 0.001). Therefore, we used Kruskal-Wallis tests to determine if there were differences in red knot and prey abundances (i.e., crustacean, blue mussel, coquina clam, miscellaneous prey, all prey) by tide (i.e., high, falling, low, rising).

**Spatial tidal sampling of prey.** Shapiro-Wilk normality tests indicated that all prey abundance data during spatial tidal sampling were not normally-distributed on peat (W = 0.42, p < 0.001) and sand (W = 0.86, p < 0.001). Therefore, we used Kruskal-Wallis and Dunn tests to determine if there were differences in prey numbers (i.e., crustacean, blue mussel, coquina clam, miscellaneous prey, all prey) by distance to central sampling point (0 meters, 5 meters, 10 meters) throughout the tidal cycle.

For all data analyses described above, we used Program R packages base, stats, dplyr, glmmTMB, lme4, and dunn.test for the analyses described above [82–86, R Version 3.4.1, www.r-project.org, accessed 6 May 2019–19 February 2020]. All tests were performed at α = 0.05 probability.

## Results

### Early migration period

We collected and analyzed 457 core samples on Virginia's barrier islands during early migration period from 2008–2018, except 2010, with a mean of 46 points per year (range = 39–61). Mean red knot flock size per sampling point was 7 (SE = 0.33). Compared to unused sites on peat banks during early migration period, red knots used sites with more crustaceans (W = 15,229, p < 0.001), blue mussels (W = 15,322, p < 0.001), miscellaneous prey (W = 13,822, p < 0.001), and all prey (W = 13,128, p < 0.001; Fig 2A).

Within the barrier islands' peat substrate during early migration, crustaceans ($\bar{x}$ = 17 173/m$^2$, SE = 2443.73) were the most abundant prey item, followed by blue mussels ($\bar{x}$ = 13 166/m$^2$, SE = 1482.90). Coquina clams were the least abundant prey ($\bar{x}$ = 119.86/m$^2$, SE = 25.18) and were only present in 21% of core samples. Miscellaneous prey, while not the most abundant prey ($\bar{x}$ = 2401/m$^2$, SE = 169.16), were the most likely to be present in core samples, with 79% of samples containing miscellaneous prey. Red knots did not use 65% of sampled peat substrate locations during early migration, while 9% of samples contained no prey. Crustacean, blue mussel, miscellaneous prey, and all prey abundances were greater on peat banks sampled during the early migration period than combined sand and peat substrates sampled during the peak migration period (S1 Table). Miscellaneous prey during early migration were comprised of angel wing clams (63%) and other organisms (37%).

**Early migration period–red knot models.** The top model containing crustaceans, coquina clams, miscellaneous prey, and tide (falling, low, rising) best explained the variation in red knot presence and flock size on peat banks during early migration (AIC$_c$ weight = 0.30; Tables 1 and S4). As the number of miscellaneous prey increased (β = -2.15, SE = 0.80), and as the tide transitioned from rising to low (β = -1.53, SE = 0.57), the probability of a zero count

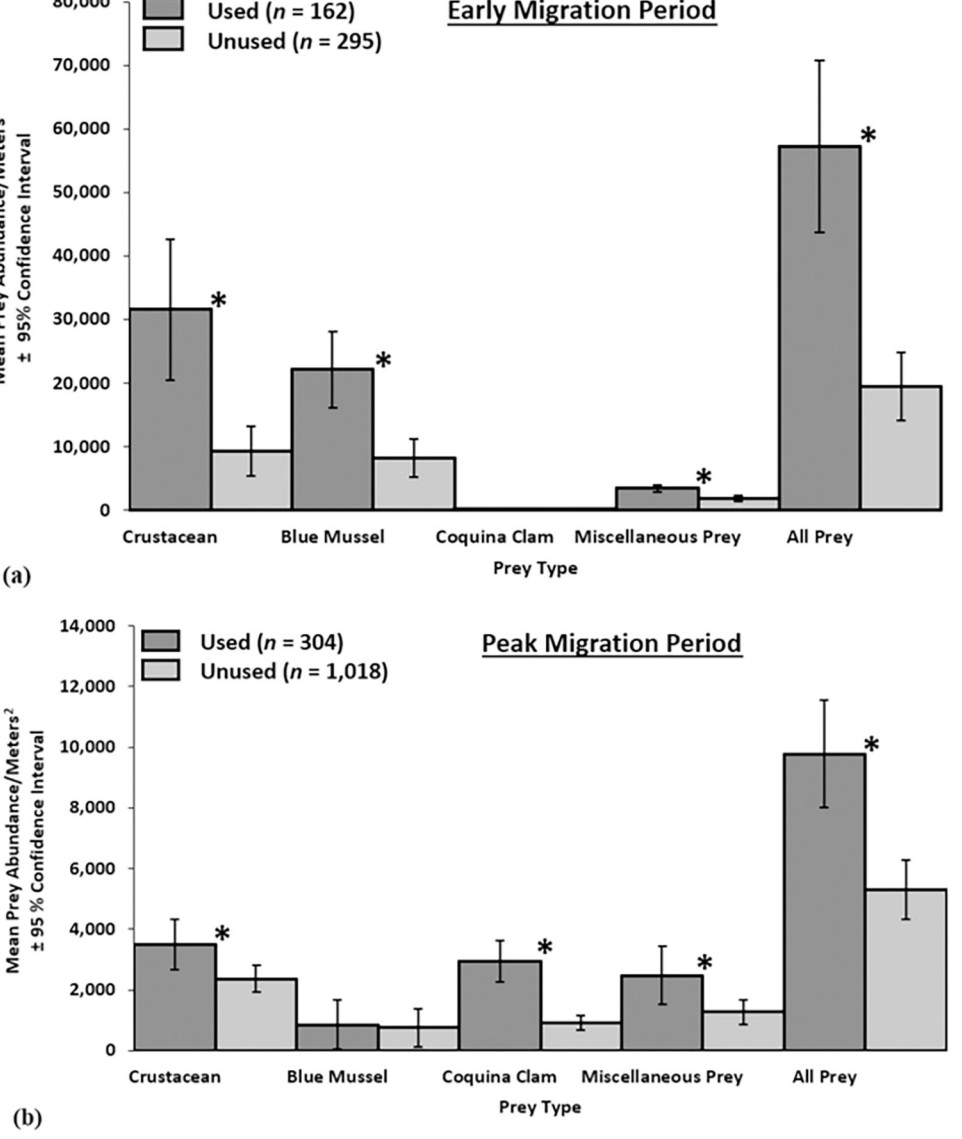

**Fig 2. Mean abundance (organisms/m²) of crustaceans, blue mussels, coquina clams, miscellaneous prey, and all prey captured in 10 cm diameter x 3.5 cm deep cores in sites used and unused by red knots.** (a) Mean abundance of organisms on peat substrate early in red knot migration (May 14–20, 2008–2018; $n$ = 457; 'early') and (b) Mean abundance of organisms on sand and peat substrate at the approximate peak of red knot migration (May 21–27, 2007–2018; $n$ = 1,322; 'peak'), Virginia's barrier islands. * Indicates a difference between used and unused sites ($p < 0.05$) based on Wilcoxon rank sum tests with Bonferroni correction.

(red knot absence) decreased. As the number of crustaceans increased, red knot flock size increased ($\beta$ = 0.23, SE = 0.11; Table 2).

**Early migration period–prey models.** Tide and mean daily water temperature best explained the variation in crustacean (AIC$_c$ weight = 0.61), coquina clam (AIC$_c$ weight = 0.71), and miscellaneous prey (AIC$_c$ weight = 0.71) abundances on peat banks during early migration (Tables 3 and S5). Crustacean, coquina clam, and miscellaneous prey abundances were highest at low tide (Table 4). As the mean daily water temperature increased, crustacean and miscellaneous prey abundances increased, while coquina clam abundance decreased (Table 4). One other model was supported for each prey type ($\Delta$AIC$_c$ < 4; Tables 3 and S5); however, the

**Table 1. Zero-inflated negative binomial mixed-effects regression models predicting red knot presence and flock size on peat substrate early in red knot migration (May 14–20, 2008–2018, except 2010; $n$ = 457; 'early'), and on sand and peat substrates at the approximate peak of red knot migration (May 21–27, 2007–2018; $n$ = 1,322; 'peak'), Virginia's barrier islands.** All models had the same covariates on both the zero-inflated and count processes and contained "Island" and "Year" as random effects.

| Period | Model | DF [a] | AIC_c [b] | ΔAICc [c] | $w_i$ [d] | LL [e] | GOF [f] |
|---|---|---|---|---|---|---|---|
| Early | Coquina Clam + Crustacean + Miscellaneous Prey + Tide | 17 | 1,743.25 | 0.00 | 0.30 | -853.93 | 0.08 |
| | Crustacean + Miscellaneous Prey [g] | 11 | 1,744.50 | 1.25 | 0.16 | -860.96 | 0.07 |
| | All Prey [h] + Tide | 13 | 1,745.28 | 2.03 | 0.11 | -859.23 | 0.07 |
| | Coquina Clam + Blue Mussel + Crustacean + Miscellaneous Prey [g] + Tide | 19 | 1,745.46 | 2.21 | 0.10 | -852.86 | 0.08 |
| | Blue Mussel + Crustacean + Miscellaneous Prey [g] | 13 | 1,745.93 | 2.68 | 0.08 | -859.55 | 0.07 |
| | Crustacean + Miscellaneous Prey [g] + Distance to Roost | 13 | 1,746.00 | 2.75 | 0.08 | -859.59 | 0.07 |
| | All Prey [h] + Distance to Roost + Tide | 15 | 1,746.68 | 3.43 | 0.05 | -857.79 | 0.07 |
| Peak | Coquina Clam + Blue Mussel + Crustacean + TDF Count [i] | 15 | 4,103.66 | 0.00 | 0.24 | -2,036.65 | 0.07 |
| | Coquina Clam + Blue Mussel + Crustacean + Miscellaneous Prey [g] + Distance to Roost + TDF Count [i] + Tide | 25 | 4,104.14 | 0.48 | 0.19 | -2,026.57 | 0.08 |
| | Coquina Clam + Blue Mussel + Crustacean + Miscellaneous Prey [g] + TDF Count [i] | 17 | 4,104.21 | 0.55 | 0.18 | -2,034.87 | 0.07 |
| | Coquina Clam + Blue Mussel + TDF Count [i] | 13 | 4,104.88 | 1.22 | 0.13 | -2,039.30 | 0.07 |
| | Coquina Clam + Blue Mussel + Distance to Roost | 13 | 4,106.03 | 2.37 | 0.07 | -2,039.88 | 0.07 |
| | Coquina Clam + Blue Mussel + Crustacean + Distance to Roost | 15 | 4,106.20 | 2.54 | 0.07 | -2,037.92 | 0.07 |
| | Coquina Clam + Blue Mussel + Crustacean + Miscellaneous Prey [g] + Distance to Roost | 17 | 4,106.70 | 3.04 | 0.05 | -2,036.12 | 0.07 |

[a] DF = Degrees of freedom.

[b] AIC_c = Akaike's Information Criterion corrected for sample size.

[c] ΔAIC_c = Difference between a model's AIC and that of the best fitting model.

[d] $w_i$ [d] = Akaike model weight.

[e] LL = Log-Likelihood.

[f] GOF = Goodness of fit = [log-likelihood(null model)–log-likelihood(model)]/log-likelihood(null model).

[g] Miscellaneous Prey = Sum of horseshoe crab eggs (*Limulus polyphemus*), angel wing clams (*Cyrtopleura costata*), and other organisms (e.g., insect larvae, snails, worms).

[h] All Prey = Sum of coquina clams + blue mussels + crustaceans + miscellaneous prey.

[i] TDF Count = Tierra del Fuego Count = Counts of red knots using Tierra del Fuego boreal wintering grounds by year (i.e., as an index for the total number of red knots in the flyway).

second ranked models contained an additional parameter, suggesting that the additional parameter (island type) was uninformative; thus, we only considered the most parsimonious model as supported for each prey type.

## Peak migration period

We collected and analyzed 1,322 samples on Virginia's barrier islands during the peak migration period ($n$ = 71 peat samples, $n$ = 1,251 sand) from 2007–2018, with a mean of 110 points per year (range = 93–129; S1 Table). Predicted numbers of red knots in Virginia did not vary between expansion methods over the study's duration (paired t-test; t = -0.47, df = 11, p = 0.65), and thus we present results on the ZINB Model Expansion, as it better accounts for the non-normal distribution of the raw count data. On average, we predicted that 7,175 (SD = 2,869) red knots used Virginia's barrier islands during peak spring migration each year. Peak red knot count numbers in Virginia were highest in 2012 (11,959) and lowest in 2014 (2,857; Fig 3); peak red knot counts in Virginia showed no linear trend over time (t = 1.04, df = 10, p = 0.32), and there were no differences in mean red knot peak counts by year (Freidman chi-squared = 18.01, df = 11, p = 0.08). There was also no correlation between the peak counts of red knots in Virginia and the Delaware Bay (t = 0.62, df = 10, p = 0.55).

**Table 2. Parameter estimates (β) from the most parsimonious zero-inflated negative binomial mixed-effect regression models predicting red knot presence (zero-inflated process) and flock size (count process) on peat substrate early in red knot migration (May 14–20, 2008–2018, except 2010; *n* = 457; 'early'), and on sand and peat substrates near the peak of red knot migration (May 21–27, 2007–2018; *n* = 1,322; 'peak'), Virginia's barrier islands.** All models contained "Island" and "Year" as random effects.

| Period | Model Process | Covariate | β [a] | SE [b] | LCI [c] | UCI [d] | z value | Pr(>\|z\|) [e] | Significance [f] |
|---|---|---|---|---|---|---|---|---|---|
| Early | Zero-Inflated | Intercept | 1.62 | 0.92 | -1.19 | 2.42 | 0.67 | 0.51 | |
| | | Coquina Clam | -0.21 | 0.42 | -1.02 | 0.61 | -0.50 | 0.62 | |
| | | Crustacean | -1.13 | 1.10 | -3.29 | 1.03 | -1.03 | 0.30 | |
| | | Miscellaneous Prey | -2.15 | 0.80 | -3.72 | -0.57 | -2.67 | 0.01 | * |
| | | Falling Tide | -1.03 | 0.64 | -2.28 | 0.23 | -1.60 | 0.11 | |
| | | Low Tide | -1.53 | 0.57 | -2.64 | -0.42 | -2.70 | 0.01 | * |
| | Count | Intercept | 1.62 | 0.59 | 0.46 | 2.77 | 2.74 | 0.01 | * |
| | | Coquina Clam | -0.02 | 0.11 | -0.24 | 0.20 | -0.17 | 0.87 | |
| | | Crustacean | 0.23 | 0.11 | 0.02 | 0.44 | 2.10 | 0.04 | * |
| | | Miscellaneous Prey | 0.15 | 0.14 | -0.13 | 0.43 | 1.04 | 0.30 | |
| | | Falling Tide | 0.16 | 0.50 | -0.81 | 1.13 | 0.32 | 0.75 | |
| | | Low Tide | 0.67 | 0.42 | -0.16 | 1.49 | 1.59 | 0.11 | |
| Peak | Zero-Inflated | Intercept | 0.05 | 0.51 | -0.95 | 1.04 | 0.09 | 0.93 | |
| | | Blue Mussel | -8.87 | 4.11 | -16.92 | -0.81 | -2.16 | 0.03 | * |
| | | Coquina Clam | -3.08 | 0.69 | -4.42 | -1.73 | -4.49 | <0.001 | * |
| | | Crustacean | -0.27 | 0.17 | -0.62 | 0.07 | -1.56 | 0.12 | |
| | | TDF Count [h] | -0.49 | 0.17 | -0.82 | -0.16 | -2.91 | 0.004 | * |
| | Count | Intercept | 3.16 | 0.30 | 2.57 | 3.75 | 10.54 | <0.001 | * |
| | | Blue Mussel | -0.30 | 0.10 | -0.49 | -0.11 | -3.05 | 0.002 | * |
| | | Coquina Clam | 0.02 | 0.09 | -0.15 | 0.19 | 0.22 | 0.83 | |
| | | Crustacean | -0.24 | 0.09 | -0.41 | -0.07 | -2.72 | 0.01 | * |
| | | TDF Count [h] | -0.26 | 0.13 | -0.52 | -0.01 | -2.02 | 0.04 | * |

[a] β = Beta estimate.

[b] SE = Standard error.

[c] LCI = Lower 95% confidence interval.

[d] UCI = Upper 95% confidence interval.

[e] Pr(>\|z\|) = Significance level.

[f] Significance = * = p ≤ 0.05.

[g] Miscellaneous Prey = Sum of horseshoe crab eggs (*Limulus polyphemus*), angel wing clams (*Cyrtopleura costata*), and other organisms (e.g., insect larvae, snails, worms).

[h] TDF Count = Tierra del Fuego Count = Counts of red knots using Tierra del Fuego boreal wintering grounds by year (i.e., as an index for the total number of red knots in the flyway).

Peak counts of red knots as expanded from linear extrapolation of ground counts (dark gray) at randomly selected 100 m radius circles centered in the swash zone and as expanded (± 95% confidence intervals) from predictions made from a zero-inflated negative binomial mixed-effects model (Red Knot ~ Island + Year; light gray) from the same ground counts, at the approximate peak of red knot migration ('peak migration period'), May 21–27, 2007–2019, Virginia's barrier islands.

Mean red knot flock size per sampling point (x̄ = 17 red knots/point, SE = 0.46) was higher during peak migration than during the early migration period (x̄ = 7 red knots/point, SE = 0.33; S1 Table). Sites used by red knots during peak migration had more crustaceans (W = 120,030, p < 0.001), coquina clams (W = 88,004, p < 0.001), miscellaneous prey (W = 140,030, p = 0.003), and all prey (W = 95,570, p < 0.001) than unused sites (Fig 2B).

**Table 3. Most parsimonious generalized linear mixed-effects regression models (ΔAIC$_c$ < 4) predicting crustacean, coquina clam, and miscellaneous prey abundances (organisms/m$^2$) captured in 10 cm diameter x 3.5 cm deep cores on peat substrate early in red knot migration (May 14–20, 2008–2018; $n$ = 457; 'early') and on sand and peat substrate at the approximate peak of red knot migration (May 21–27, 2007–2018; $n$ = 1,322; 'peak'), Virginia's barrier islands.** All models contained "Island" and "Year" as random effects.

| Period | Prey | Model | DF[a] | AIC$_c$ [b] | ΔAIC$_c$ [c] | $w_i$[d] | LL[e] | MR2[f] | CR2[g] |
|---|---|---|---|---|---|---|---|---|---|
| Early | Crustacean | Tide + Water Temperature | 6 | 110,491.00 | 0.00 | 0.61 | -55,239.42 | 0.02 | 0.57 |
| | | Tide + Water Temperature + Island Type | 7 | 110,491.90 | 0.88 | 0.39 | -55,238.83 | 0.06 | 0.56 |
| | Coquina Clam | Tide + Water Temperature | 6 | 1,952.17 | 0.00 | 0.71 | -969.99 | 0.01 | 0.00 |
| | | Tide + Water Temperature + Island Type | 7 | 1,954.23 | 2.06 | 0.25 | -969.99 | 0.01 | 0.01 |
| | Misc. Prey[h] | Tide + Water Temperature | 6 | 8,721.57 | 0.00 | 0.71 | -4,354.69 | 0.04 | 0.69 |
| | | Tide + Water Temperature + Island Type | 7 | 8,723.40 | 1.84 | 0.29 | -4,354.58 | 0.07 | 0.66 |
| Peak | Crustacean | Tide + Substrate + Water Temperature | 8 | 61,060.87 | 0.00 | 0.56 | -30,522.38 | 0.02 | 0.22 |
| | | Tide + Substrate + Water Temperature + Island Type | 9 | 61,061.39 | 0.51 | 0.44 | -30,521.62 | 0.05 | 0.23 |
| | Coquina Clam | Tide + Substrate + Water Temperature | 8 | 35,893.06 | 0.00 | 0.69 | -17,938.47 | 0.04 | 0.26 |
| | | Tide + Substrate + Water Temperature + Island Type | 9 | 35,894.64 | 1.58 | 0.31 | -17,938.25 | 0.05 | 0.26 |
| | Blue Mussel | Tide + Substrate + Water Temperature + Island Type | 9 | 22,816.70 | 0.00 | 0.51 | -11,399.28 | 0.06 | 0.07 |
| | | Tide + Substrate + Water Temperature | 8 | 22,816.77 | 0.07 | 0.49 | -11,400.33 | . | . |

[a] DF = Degrees of freedom.

[b] AIC$_c$ = Akaike's Information Criterion corrected for sample size.

[c] ΔAIC$_c$ = Difference between a model's AIC and that of the best fitting model.

[d] $w_i$[d] = Akaike model weight.

[e] LL = Log-Likelihood.

[f] MR2 = Marginal r-squared = Considers only the variance of the fixed effects.

[g] CR2 = Conditional r-squared = Considers the variance of both fixed and random effects.

[h] Misc. Prey = Miscellaneous Prey = Sum of horseshoe crab eggs (*Limulus polyphemus*), angel wing clams (*Cyrtopleura costata*), and other organisms (e.g., insect larvae, snails, worms).

Across the barrier islands' intertidal shoreline (including both peat banks and sand), crustaceans were the most abundant prey (x̄ = 2621/m$^2$, SE = 199.65; Fig 4A) and were collected in 80% of samples (S1 Table). Blue mussels were the least abundant prey (x̄ = 778/m$^2$, SE = 259.31; Fig 4A) and were only collected in 4% of samples (S1 Table). Red knots did not use 77% of sampling locations, while 10% of sampling locations did not contain any prey (S1 Table). With both substrates included, miscellaneous prey was comprised of other organisms (96%), angel wing clams (2%), and horseshoe crab eggs (2%). When separating sites by substrate from samples collected during this peak migration period, the composition of prey differed between sand and peat substrates ($\chi^2$ = 15,793, df = 4, p < 0.001). Blue mussels (W = 68,288, p < 0.001) and miscellaneous prey (W = 57,534, p < 0.001) were more abundant on peat than on sand, while coquina clams were more abundant on sand than on peat (W = 29,832, p < 0.001; Fig 4B). When separated by substrate, miscellaneous prey were as follows: peat–other organisms (69%) and angel wing clams (31%); sand—other organisms (97%), horseshoe crab eggs (2%), and angel wing clams (1%).

**Peak migration period–red knot models.** The top ranked model containing crustacean, blue mussel, and coquina clam abundances in Virginia and boreal wintering counts of red knots in Tierra del Fuego best explained the variation in red knots in Virginia during peak migration (AIC$_c$ weight = 0.24; Tables 1 and S4). The probability of a zero count decreased with increasing blue mussel abundance (β = -8.87, SE = 4.11), coquina clam abundance (β = -3.08, SE = 0.69), and the number of red knots using Tierra del Fuego boreal wintering grounds (β = -0.49, SE = 0.17; Table 2). However, red knot flock size decreased with increasing

**Table 4. Parameter estimates (β) from the most parsimonious generalized linear mixed effects regression models predicting crustacean, coquina clam, and miscellaneous prey abundances (organisms/m²) captured in 10 cm diameter x 3.5 cm deep cores on peat substrate early in red knot migration (May 14–20, 2008–2018, except 2010; *n* = 457; 'early'), and on sand and peat substrates near the peak of red knot migration (May 21–27, 2007–2018; *n* = 1,322; 'peak'), Virginia's barrier islands.** All models contained "Island" and "Year" as random effects.

| Period | Prey | Covariate | β [a] | SE [b] | LCI [c] | UCI [d] | z value | Pr(>\|z\|) [e] | Significance [f] |
|--------|------|-----------|-------|--------|---------|---------|---------|----------------|------------------|
| Early | Crustacean | Intercept | -4.75 | 0.81 | -6.33 | -3.17 | -5.89 | <0.001 | * |
| | | Low Tide | 0.54 | 0.01 | 0.52 | 0.56 | 54.90 | <0.001 | * |
| | | Rising Tide | -0.95 | 0.02 | -1.00 | -0.91 | -41.34 | <0.001 | * |
| | | Water Temperature | 0.50 | 0.01 | 0.48 | 0.52 | 47.58 | <0.001 | * |
| | Coquina Clam | Intercept | 2.33 | 1.24 | -0.10 | 4.76 | 1.88 | 0.06 | |
| | | Low Tide | 0.82 | 0.13 | 0.57 | 1.08 | 6.29 | <0.001 | * |
| | | Rising Tide | 0.74 | 0.18 | 0.39 | 1.09 | 4.17 | <0.001 | * |
| | | Water Temperature | -0.24 | 0.08 | -0.40 | -0.08 | -2.91 | <0.001 | * |
| | Misc. Prey[g] | Intercept | -0.70 | 0.63 | -1.93 | 0.54 | -1.10 | 0.27 | * |
| | | Low Tide | 0.10 | 0.03 | 0.04 | 0.15 | 3.44 | <0.001 | * |
| | | Rising Tide | -0.18 | 0.04 | -0.26 | -0.11 | -4.65 | <0.001 | * |
| | | Water Temperature | 0.19 | 0.02 | 0.16 | 0.23 | 9.73 | <0.001 | * |
| Peak | Crustacean | Intercept | 1.47 | 0.25 | 0.99 | 1.96 | 5.97 | <0.001 | * |
| | | Falling Tide | 0.16 | 0.02 | 0.12 | 0.21 | 6.95 | <0.001 | * |
| | | Low Tide | 0.76 | 0.02 | 0.71 | 0.80 | 31.66 | <0.001 | * |
| | | Rising Tide | 0.30 | 0.02 | 0.26 | 0.35 | 12.68 | <0.001 | * |
| | | Sand Substrate | -0.90 | 0.02 | -0.95 | -0.86 | -41.90 | <0.001 | * |
| | | Water Temperature | 0.12 | 0.01 | 0.10 | 0.14 | 10.66 | <0.001 | * |
| | Coquina Clam | Intercept | 5.64 | 0.45 | 4.77 | 6.52 | 12.62 | <0.001 | * |
| | | Falling Tide | 0.84 | 0.03 | 0.78 | 0.90 | 26.61 | <0.001 | * |
| | | Low Tide | 0.83 | 0.04 | 0.76 | 0.91 | 22.93 | <0.001 | * |
| | | Rising Tide | 0.51 | 0.03 | 0.45 | 0.57 | 15.67 | <0.001 | * |
| | | Sand Substrate | 0.87 | 0.07 | 0.73 | 1.01 | 12.27 | <0.001 | * |
| | | Water Temperature | -0.34 | 0.02 | -0.37 | -0.30 | -19.14 | <0.001 | * |
| | Blue Mussel | Intercept | -13.54 | 1.60 | -16.67 | -10.40 | -8.47 | <0.001 | * |
| | | Falling Tide | 1.05 | 0.18 | 0.69 | 1.41 | 5.73 | <0.001 | * |
| | | Low Tide | 2.90 | 0.19 | 2.54 | 3.27 | 15.69 | <0.001 | * |
| | | Rising Tide | 2.21 | 0.19 | 1.83 | 2.58 | 11.67 | <0.001 | * |
| | | Sand Substrate | -1.51 | 0.03 | -1.56 | -1.46 | -55.06 | <0.001 | * |
| | | Water Temperature | 0.59 | 0.07 | 0.46 | 0.72 | 8.85 | <0.001 | * |

[a] β = Beta estimate.

[b] SE = Standard error.

[c] LCI = Lower 95% confidence interval.

[d] UCI = Upper 95% confidence interval.

[e] Pr(>\|z\|) = Significance level.

[f] Significance = * = p ≤ 0.05.

[g] Misc. Prey = Miscellaneous Prey = Sum of horseshoe crab eggs (*Limulus polyphemus*), angel wing clams (*Cyrtopleura costata*), and other organisms (e.g., insect larvae, snails, worms).

blue mussel abundance (β = -0.30, SE = 0.10), crustacean abundance (β = -0.24, SE = 0.09), and the number of red knots using Tierra del Fuego (β = -0.26, SE = 0.13; Table 2). While six other models had ΔAIC$_c$ values less than 4, the second ranked model contained ten additional parameters, suggesting that the additional parameters were uninformative (Arnold 2010; S4 Table); thus, we only considered the most parsimonious model as supported.

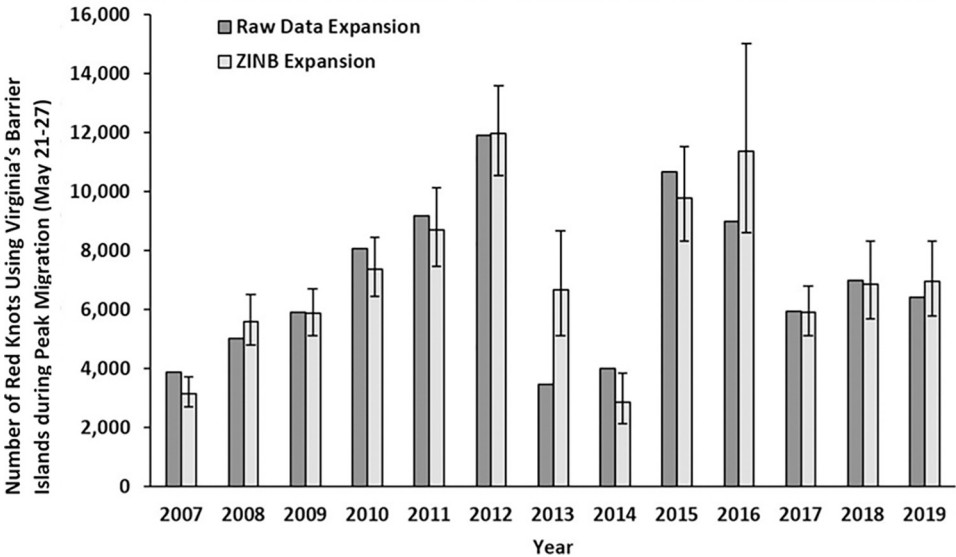

**Fig 3. Peak counts of red knots.**

**Peak migration period–prey models.** Tide, substrate, and mean daily water temperature best explained the variation in crustacean ($AIC_c$ weight = 0.56), coquina clam ($AIC_c$ weight = 0.69), and blue mussel ($AIC_c$ weight = 0.49) abundances during peak migration (Tables 3 and S5). Crustacean and blue mussel abundances were highest at low tide, while coquina clam abundances were highest at falling and rising tides. Crustaceans and blue mussels were more abundant on peat, while coquina clams were more abundant on sand. As the mean daily water temperature increased, crustacean and blue mussel abundances increased, while coquina clam abundance decreased (Table 4). While the blue mussel model with tide, substrate, mean daily water temperature, and island type ($AIC_c$ weight = 0.51; S5 Table) carried the most weight, because there was no difference in blue mussel abundance on low vs. high elevation islands, we considered the second ranked model described above as most parsimonious (Arnold 2010).

**Peat and sand tidal sampling.** Crustacean abundance varied by tide during peat tidal sampling ($\chi^2$ = 13.03, df = 2, p = 0.001), with crustacean abundance being highest at low tide. Red knot flock size ($\chi^2$ = 36.90, df = 3, p < 0.001) and coquina clam ($\chi^2$ = 19.51, df = 3, p < 0.001) and blue mussel (Kruskal-Wallis; $\chi^2$ = 13.46, df = 3, p = 0.003) abundances varied by tide during sand tidal sampling, with red knot flock size and blue mussel and coquina clam abundances being highest at rising tide.

**Tidal spatial sampling.** Only all combined prey abundance varied by distance (5 m, 10 m) from a central sampling point (0 m) during peat tidal spatial sampling conducted in 2019 (Kruskal-Wallis; $\chi^2$ = 6.90, df = 2, p = 0.03). The overall combined prey abundance differed between 0 m and 10 m, with all prey being higher at 0 m (z = 2.63, p = 0.01; Fig 5).

## Discussion

Site selection by red knots and prey availability varied by substrate and tide. While most (~ 90%) sampling locations contained prey, red knots did not use 64–77% of all sampling locations, suggesting that red knots use a small proportion of habitat containing prey at any given time in Virginia. The sites that red knots used contained higher prey abundances than unused sites, supporting the work of previous studies [31,32,41,87,88]. These relationships suggest that

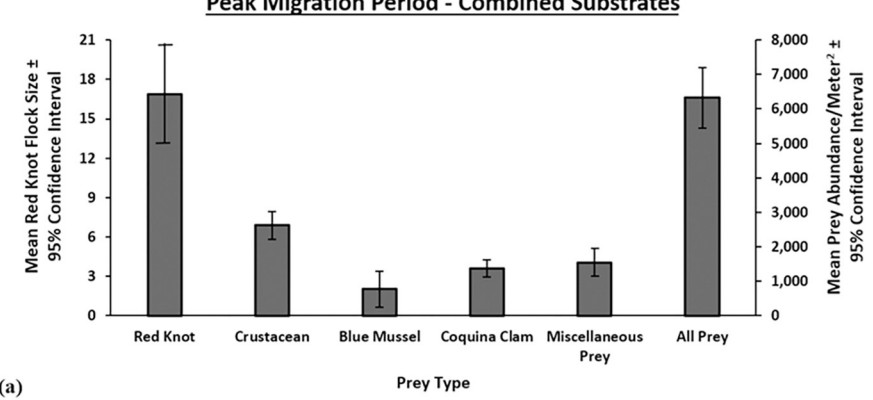

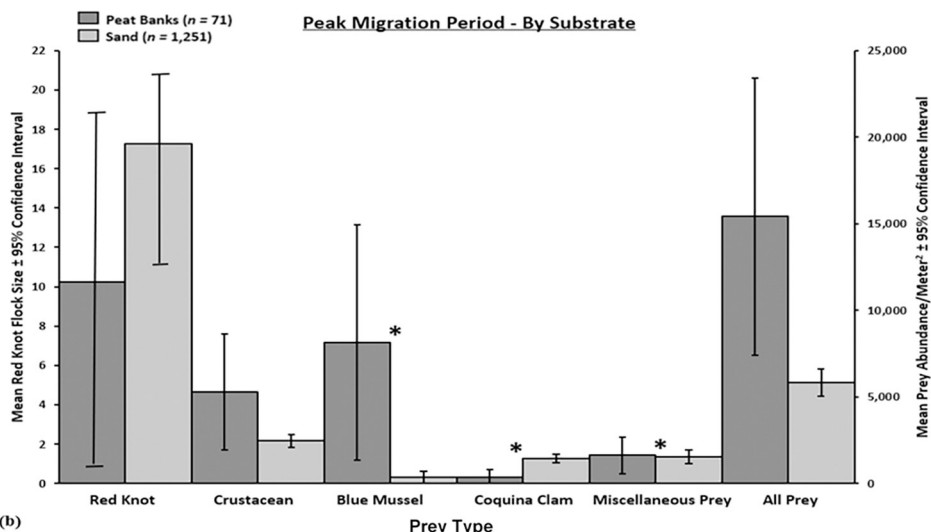

**Fig 4. Red knot flock size and mean prey abundance for combined substrates and individually by peat bank and sand substrate for samples collected during red knot peak migration (21–27 May).** Mean red knot flock size per 100 m radius semicircle on the water line and mean abundance (organisms/m$^2$ ± 95% CI) of crustaceans, blue mussels, coquina clams, miscellaneous prey, and all prey captured in 10 cm diameter x 3.5 cm deep cores on (a) peat and sand substrates combined (n = 1,322) and on (b) peat banks (n = 71) and sand (n = 1,251) separately at the approximate peak of red knot migration ('peak migration period'), May 21–27, 2007–2018, Virginia's barrier islands. * Indicates a difference (p < 0.05) in red knot flock size and prey abundances between peat and sand substrates based on Wilcoxon rank sum tests with Bonferroni correction. Note: y-axis scales differ between (a) and (b).

red knots decrease energy expenditure and maximize foraging efficiency by foraging in locations that are most profitable (i.e., those that provide the highest abundance of prey in the shortest period [41,89]). Prior work in other systems has also consistently demonstrated that red knots likely feed on prey that are most digestible [90–94]. For example, in the Dutch Wadden Sea site, which closely resembles the intertidal habitat of Virginia's barrier islands, red knots selected juvenile edible cockles (*Cerastoderma edule*) that had thinner shells and proportionately high flesh content to maximize energy intake rates and reduce processing time (i.e., shell digestion [95,96]). Other studies have also demonstrated that shorebirds such as sanderling (*Calidris alba*), Eurasian oystercatchers (*Haematopus ostralegus*), and dunlin (*Calidris alpina*) select prey that maximize their caloric intake [97–99]. Additional study into the energy content and digestibility of all prey available to red knots in Virginia would help further clarify the relationship among red knots, prey, and habitat selection.

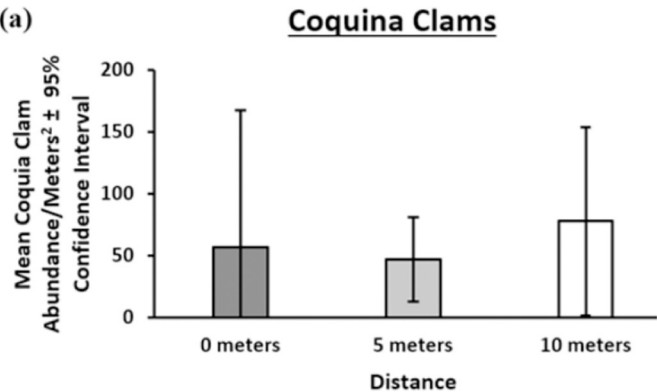

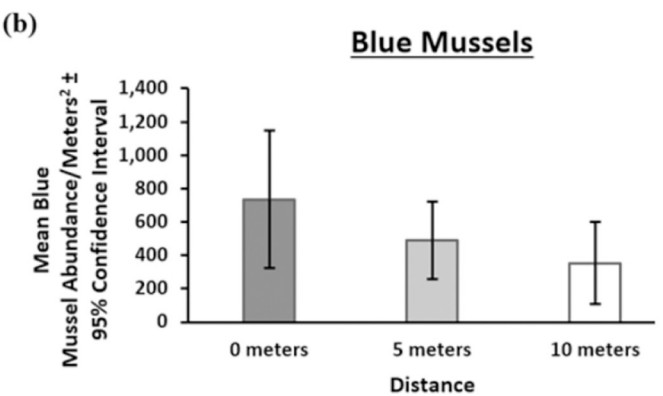

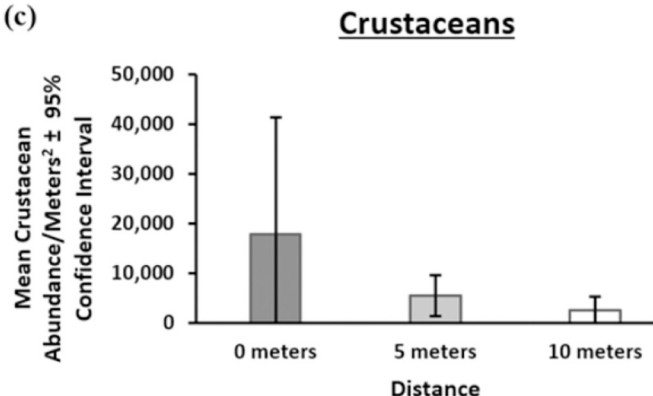

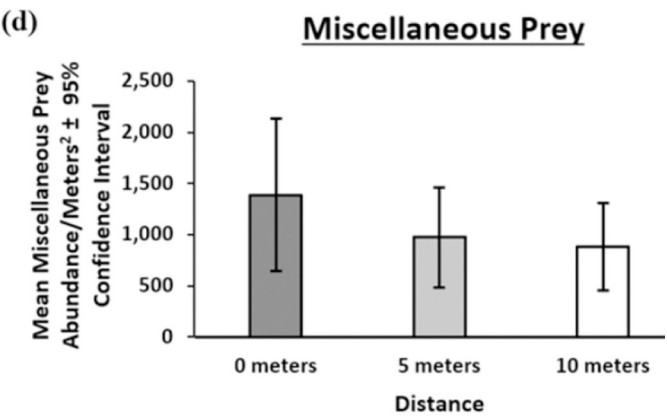

**Fig 5. Mean prey abundances at distances from 0 m to 10 m from the main sample point during tidal spatial sampling on peat banks.** Mean (a) coquina clam, (b) blue mussel, (c) crustacean, (d) miscellaneous prey, and (e) all prey abundances (organisms/m$^2$) captured in 10 cm diameter x 3.5 cm deep cores on peat banks during peat tidal spatial sampling, May 17, 2019, Myrtle Island, Virginia's barrier islands. Distance represents the distance from the center sampling point (0 meters). * Kruskal-Wallis and Dunn tests found a difference (p < 0.05) in all prey abundance between 0 m and 10 m.

While the probability of at least one red knot was greater at sites with more prey, red knot flock size did not consistently relate to prey abundance. Most prey abundances (i.e., blue mussels, crustaceans, miscellaneous, and all prey) were highest during early migration period when only peat substrates were sampled due to logistical constraints, while red knot flock size was highest during peak migration period when more samples fell on sand than peat substrate as sampling was done irrespective of tide state. This relationship is likely an artifact of the different sampling approaches used during early migration, when we only sampled random points on peat substrate, and peak migration, when we sampled random points on both peat and sand, as we found prey abundance to be higher in peat than in sand. Therefore, direct comparisons of red knot and prey numbers between early and peak migration periods, without accounting for substrate type, cannot be made under our current sampling design. If logistical constraints could be released so that we could equally sample both peat and sand substrates during both the early and peak migration periods, then more direct comparisons between red knot numbers and prey abundances on the two substrate types could be made.

While crustaceans positively influenced red knot flock size during the early migration period, crustaceans and blue mussels were negatively related to red knot flock size during the peak migration period, likely because these two prey species live predominantly in peat substrate. Although these red knot flock size—prey relationships may seem counterintuitive, as we would expect larger red knot flock sites at sites with more prey, the discrepancy may be influenced by five factors. First, we did not sample red knots at high tide or on sand substrate during the early migration period, which may have negatively biased our flock size estimates during early migration by not counting birds that roost, typically on sand, in large numbers at high tide [28,100]. Second, some of the relationships between red knot flock size and prey during peak migration (i.e., the negative correlation between red knot flock size and blue mussels and crustaceans) may be habitat correlates, related to the propensity of some prey to settle on peat substrate that were sampled less during peak migration than early migration, rather than truly negative relationships [32]. Third, flock size alone is not directly related to the overall number of birds using the islands. Fourth, prey may have been depleted at some sites during peak migration before red knots were counted [32,33], and fifth, site suitability varies in the intertidal zone. For example, some peat banks are located far above the low-tide line and thus become dry throughout the tidal cycle. These peat banks do not support the same abundances of prey as peat that remains saturated throughout the tidal cycle. While the abundance of prey may not consistently affect flock size within a sampling period, consistent prey availability across the migration window may affect overall red knot abundance on the islands [8]. For example, we designed our study to conduct sampling during early migration (May 14–20) and peak migration (May 21–27) using historical data that suggested these two weeks corresponded to early and peak migration for red knots using Virginia and Delaware Bay staging sites [26,29,101]; thus, excluding any effects of prey, we would expect more red knots to use Virginia's barrier islands during peak migration period than during early migration period, regardless of flock size estimates.

Consideration of the multiple drivers of red knot flocking behavior also offers insight into our ability to explain red knot presence better than red knot flock size at a given point in

Virginia. For example, Folmer et al. [102] found that the predictive ability of resource-related (e.g., prey) models on the spatial distribution of foraging shorebirds decreased with the tendency of a species to flock. As social birds [95,103], red knots prefer to forage in non-random groupings (i.e., flocks) and therefore likely base their foraging decisions at least partly on conspecific attraction and perceptions [102,104]. These conspecific interactions may help red knots determine food availability and the presence of predators at a given site [102]. Additionally, peregrine falcons (*Falco peregrinus*), predators of red knots, live on Virginia's barrier islands, where shorebirds provide an estimated 52% of the peregrine falcon's diet [105,106]. Thus, conspecific interactions may also cause red knots to select a site simply because others do so to maximize predator detection, regardless of prey availability ("many eyes hypothesis" [107,108]).

During peak migration, red knot presence and flock size were also influenced by boreal wintering red knot counts in Tierra del Fuego, which were used here as an index of red knots available in the flyway to stage in Virginia. The positive relationship between Tierra del Fuego counts and red knot presence suggests that the more birds in the boreal wintering grounds, and presumably the flyway, the more sites in Virginia will be occupied in a given year. The negative relationship between Tierra del Fuego boreal wintering counts and flock size during peak migration period in Virginia suggests that red knots occupy more sites in Virginia, but in smaller flocks, when Tierra del Fuego counts are higher. Birds may separate into smaller flocks to exploit more foraging sites, while simultaneously decreasing competition within a site [109]. However, that prediction is in contrast to numerous other studies that suggest that larger flock sizes may help animals determine site suitability based on the abundance of available prey [88,110–112]. Further assessment of the relative contribution of other boreal wintering populations to the Virginia staging site, as Smith [69] began to detail, could help elucidate this perplexing negative relationship between red knot numbers in Tierra del Fuego and flock size in Virginia. It is possible that the total numbers of red knots coming to Virginia from other boreal wintering sites in the flyway varies annually such that counts of red knots in Tierra del Fuego cannot fully explain both the presence and flock size of red knots in Virginia.

Blue mussel and miscellaneous prey densities were higher in peat than in sand substrate. Some prey, like blue mussels, require a substrate (e.g., peat) on which to attach [113]. Other prey that do not attach to the substrate may prefer to live in peat to more easily hide from predators in dense vegetation and/or because they consume various decaying plant and animal material found in the peat banks [114–117]. In contrast, coquina clam density was higher in sand than in peat. Coquina clams prefer sand substrate that enables them to migrate both vertically and horizontally across the shoreline throughout the tidal cycle, primarily migrating shoreward during rising tides, seaward during falling tides, and remaining idle during low and high tides [118,119]. These migrations likely decrease predation risk by keeping the clams mobile, by preventing them from becoming stranded at high tide, and also by increasing the clams' foraging efficiency through reduced risk of resource depletion within one area [120,121]. However, these prey-substrate relationships do not account for the abundance of prey across the entire intertidal zone, only the abundance of prey found in collected core samples. Because peat banks comprise only ~ 6% of the intertidal zone each year, while sand comprises the remainder of the intertidal zone [41,55], the cumulative abundance of prey in sand is likely much greater than the abundance of prey in peat on the entire barrier island chain. However, both substrates are important in supporting different types of prey and provide prey at different times in the tidal cycle. For example, blue mussels and crustaceans were captured in the highest densities within two hours of low tide, corresponding to the time of greatest peat substrate exposure, and coquina clams generally were most abundant during the falling and rising tides when clams engage in tidal migrations. Red knot flock size was highest at low tide

during early migration when only peat was sampled. Red knot flock size and prey abundances were also generally highest around the rising tide during sand tidal sampling. While prey varied by substrate and tide, and despite other studies finding that invertebrate prey abundance is highly variable across space [122–124], the only difference in prey abundances within 10 meters of a central sampling point occurred when all prey were combined during peat tidal sampling. The lack of consistent differences over space suggests that larger scale covariates (e.g., ocean temperature) may affect prey abundance and distribution throughout the tidal cycle or that the spacing we selected (5 m and 10 m from a central point) were inconsistent with prey spatial variation.

Crustaceans were abundant on both peat and sand. Heller [42] found that despite the high abundance of crustaceans across the intertidal landscape of Virginia's barrier islands, red knots selected crustaceans less than expected given their availability. If ocean temperatures continue to warm [125,126], causing further range contraction and decline in the abundance of blue mussels in Virginia [42], red knots may need to rely more heavily on abundant crustaceans or other prey that we grouped in the miscellaneous category (horseshoe crab eggs, angel wing clams, and other organisms (e.g., insect larvae, snails, worms)) in Virginia. However, coquina clams may become larger and/or more abundant in Virginia due to ocean warming, as they grow faster and mature earlier in warmer water [127]. Previous studies estimating the caloric value of crustaceans (*Corophium sp.*) consumed by redshank (*Tringa tetanus*), a shorebird of similar size to red knots, found that crustaceans contained on average 18.20 kJ/g ash-free dry weight (AFDW, [128]). In another study of the energetic content of shorebird prey in the Dutch Wadden Sea, Zwarts and Wanink [129] found that the same species of blue mussels consumed by red knots in Virginia, *Mytilus edulis*, had the highest energetic content among ten tidal invertebrates measured, with blue mussels having on average about 23.4 kJ/g AFDW. These data demonstrate that crustaceans are less energetically dense than bivalves although still a potentially energetically-rich food resource for red knots; however, red knots' will likely continue to select bivalves as long as they are available given their high energetic content and the tactile foraging adaptation of red knots that enables them to efficiently capture bivalve prey [130,131].

The peak counts of red knots using Virginia's barrier islands were variable over the study's duration, though no positive or negative linear trend existed. We acknowledge that our approach to predict the abundance of red knots in Virginia during peak migration was imperfect, but we minimized potential biases associated with these assumptions by sampling red knots during peak migration (May 21–27) after most have arrived on the islands and when they tend to stay for at least 2 weeks [29], by having only highly trained observers identify and count red knots, and by staying at least 100 meters away from flocks to prevent dispersal. Additionally, our average peak prediction of 7,175 red knots per year (2007–2018) was close to the average peak red knot count obtained using aerial surveys from 2007–2014 (x̄ = 6,788 red knots per year [23,125,126]. Our average peak count of red knots each year (x̄ = 6,521 red knots per year; 2007–2014) was also close to aerial peak count estimates when we averaged only our ground count predictions for the years during which aerial counts were also made (x̄ = 6,788 red knots per year, 2007–2014 [132]). We are further encouraged by our ability to detect trends in red knot numbers using these ground count predictions as our observed decrease in peak counts of red knots from 11,959 in 2012 to 6,670 and 2,857 in 2013 and 2014 respectively also coincided to declines observed via aerial surveys in those years (8,482 in 2012 to 6,200 and 5,547 in 2013 and 2014 respectively, [133]). Thus, our ground count estimates of red knot numbers and the annual aerial flight counts of red knots during the years of 2007–2014 [132,133] showed a similar mean peak count and similar trends in numbers despite the varied methods and assumptions of each.

Cumulatively, our study suggests that the number of red knots in Virginia result from complex interactions both on the staging site and beyond. Previous studies of red knot population trends (including *Calidris canutus rufa* and *Calidris canutus canutus*) demonstrated that red knot populations often cycle within 3 to 4-year periods. Years of high lemming population sizes resulted in high reproductive output of red knots and consequently higher red knot numbers the following 1–3 years [35,134,135]. However, in the late 1990s, the lemming cycle in Europe was altered [136–138]. It is less clear whether lemming cycles changed in North America; however, if they did, any lemming cycle cessation or alteration may have contributed to the *rufa* red knot's population decline in the early 2000s [35]. Predicted red knot numbers in Virginia during this study did not visibly follow the same 3–4 year cycle as seen in Sutherland [135] and Fraser et al. [35], but there were repeated highs and lows. An examination of potential cyclic patterns may be warranted if long-term modelling continues to show variable trends over time.

The annual fluctuations in the red knot population that migrates through the mid-Atlantic region [35,133] and the proportion of red knots that use the Virginia staging site may be related to the quality of Delaware Bay and Virginia staging site habitat within a given year. Our long-term findings regarding the relationship between red knots and their prey in Virginia only explain some of the variation in red knot site use and flock size. Because there is no evidence of extreme red knot population fluctuations since their decline in the mid-1990s (this study, [22,23]), and because prey abundance varies over space and time, we speculate that any variation in the number of red knots using the Virginia barrier islands is at least partly due to the abundance and quality of prey at other locations across the annual cycle, including other staging and stopover areas throughout the Western hemisphere. The factors that affect the presence, flock size, and abundances of long-distance migrants using migratory staging sites grounds are not straightforward, as factors across their boreal wintering, breeding, and other staging grounds likely affect birds year-round [16,17]. Thus, additional studies that link potentially relevant variables across each area within the range that red knots use during their life-cycle are warranted to best design successful management practices and develop conservation priorities range-wide [139].

## Conclusions

Red knots historically frequented a larger region in coastal North America from Florida to Massachusetts as migratory stopover habitat than they do today [12,27]. The largest of the red knot spring migratory staging sites now are primarily found on the Delaware Bay and Virginia's barrier islands (~ 50–70% of the annual spring migration population [20,23,26,29]). Because these staging grounds support high percentages of the entire migratory population, any deviation in the historic norm of habitat and prey availabilities may have lasting population-wide implications for red knots and other long-distance migratory shorebirds [140].

Our research suggests that although blue mussels and coquina clams are important prey resources for red knots, counts of red knots in boreal wintering grounds and counts of other types of prey, such as crustaceans, may also be important predictors of red knot presence and flock size in Virginia. To continue maximizing the availability of red knot prey across the tidal cycle, and in particular the availability of blue mussel prey which requires peat bank substrate to settle in high densities, ongoing management on Virginia's barrier islands that discourages beach stabilization and nourishment projects and allows the natural processes of overwash and island transgression should continue. Beach nourishment buries invertebrate prey that live within the top layers of sand and peat, causing prey mortality, altered prey community assemblages, and/or a reduction in foraging shorebirds' ability to access prey [141–147]. Beach

stabilization and nourishment stall coastal shoreline erosion and are often used on barrier islands to prevent island transgression [142,148,149]. However, peat banks cannot form when islands are unable to transgress over themselves onto back-side marsh [47,141]. Therefore, beaches that are nourished or otherwise managed to prevent erosion generally lack peat banks. Over time, the loss of peat bank habitat would likely decrease the abundance of peat-reliant red knot prey species, like blue mussels and crustaceans.

## Supporting information

**S1 Fig. Sample points for 2018, taken from [51], as an example for other years on the Virginia Coast Reserve Long Term Ecological Research Site on 11 barrier islands off the Eastern Shore of Virginia.** We sampled red knots and invertebrate prey at randomly generated points, each separated by at least 200m, on 11 barrier islands. Sample points collected during the early migration period (peat substrate only, May 14–20, 2018) are shown in dark blue whereas sample points during the peak migration period (peat and sand substrate, May 21–27, 2018) are shown in light blue on the images. From north to south, samples were collected on Assawoman Island, Metompkin Island, Cedar Island, Parramore Island, Hog Island, Cobb Island, Wreck Island, Ship Shoal Island, Mink and Myrtle Island, Smith Island and Fisherman Island. Basemap aerial island imagery from 2021 was taken from the United States Department of Agriculture's National Agriculture Imagery Program (NAIP), available at [50] from the Aerial Photography Field Office. The geographic coordinates and details on all sample points from 2007–2018 can be found in [51].
(TIF)

**S1 Table. Mean red knot flock size per 100 m radius semicircle on the water line and mean abundance (organisms/m$^2$ ± SE) of coquina clams, blue mussels, crustaceans, miscellaneous prey, and all prey captured in 10 cm diameter x 3.5 cm deep cores on peat banks early in red knot migration (May 14–20, 2008–2018; $n$ = 457; 'early') and on sand and peat banks at the approximate peak of red knot migration (May 21–27, 2007–2018; $n$ = 1,322; 'peak'), Virginia's barrier islands.**
(DOCX)

**S2 Table. Mean red knot flock size and standard errors per 100 m radius semicircle on the water line on peat banks early in red knot migration (May 14–20, 2008–2018; $n$ = 457; 'early') and on sand and peat banks at the approximate peak of red knot migration (May 21–27, 2007–2018; $n$ = 1,322; 'peak'), Virginia's barrier islands.**
(DOCX)

**S3 Table. Pearson correlation coefficients for all continuous covariates on peat banks early in red knot migration (May 14–20, 2008–2018; $n$ = 457; 'early') and on sand and peat banks at the approximate peak of red knot migration (May 21–27, 2007–2018; $n$ = 1,322; 'peak'), Virginia's barrier islands.** Highly correlated covariate combinations are represented by values > 0.7 and < -0.7 (Booth et al. 1994, Anderson et al. 2001).
(DOCX)

**S4 Table. Full model sets for zero-inflated negative binomial mixed-effects regression models predicting red knot presence and flock size on peat banks early in red knot migration (May 14–20, 2008–2018, except 2010; $n$ = 457; 'early'), and on sand and peat substrates at the approximate peak of red knot migration (May 21–27, 2007–2018; $n$ = 1,322; 'peak'), Virginia's barrier islands.** All models had the same covariates on both the zero-inflated and

count processes and contained "Island" and "Year" as random effects.
(DOCX)

**S5 Table. Full model sets for generalized linear mixed-effects regression models predicting crustacean, coquina clam, and miscellaneous prey abundances (organisms/m$^2$) captured in 10 cm diameter x 3.5 cm deep cores on peat banks early in red knot migration (May 14–20, 2008–2018; $n$ = 457; 'early') and on sand and peat banks at the approximate peak of red knot migration (May 21–27, 2007–2018; $n$ = 1,322; 'peak'), Virginia's barrier islands.** All models contained "Island" and "Year" as random effects.
(DOCX)

## Acknowledgments

Field work was conducted by N. Avissar, M. Brinckman, V. D'Amico, D. Fraser, B. Gerber, M. M. Griffin, K. Guerena, C. C. Kontos, A. J. Macan, B. Marine, B. McLaughlin, A. C. Montgomery, G. Moore, R. Releyvich, S. Ritter, T. St. Clair, K. Tatu, A. Lipford, A. Grimaudo, T. Pirault, C. Rowe, H. Glass, A. Bolivar, and H. Dixon. Laboratory assistance was provided by K. Ballagh, C. Hitchens, K. Minton, R. Slack, J. Stiles, A. Lipford, A. Grimaudo. T. Pirault, A. Scott, T. Russel, H. Wojtysiak, E. Reasor, G. Martin, C. Linkous, and H. Dixon. J. Walters assisted in study design. D. Fraser managed marine operations. Kristy Lapenta assisted in making figures for the revised manuscript. We thank Alex Wilke and Barry Truitt (The Nature Conservancy), Ruth Boettcher and Shirl Dressler (Virginia Department of Game and Inland Fisheries), Dot Field and Rick Myers (Virginia Department of Conservation and Recreation's Natural Heritage Program), Kevin Holcomb, Pam Denmon, and Rob Leffel (United States Fish and Wildlife Service), and John Bill and Rob O'Reilly (Commonwealth of Virginia Marine Resources Commission) for their logistical, permitting, and biological support and knowledge over the course of this study. We thank Academic Editor Dr. Vitor Hugo Rodrigues Paiva and 3 anonymous reviewers for their constructive feedback on this manuscript.

## Author Contributions

**Conceptualization:** Erin L. Heller, Sarah M. Karpanty, Jonathan B. Cohen, Barry R. Truitt, James D. Fraser.

**Data curation:** Erin L. Heller, Sarah M. Karpanty, Jonathan B. Cohen, Shannon J. Ritter, James D. Fraser.

**Formal analysis:** Erin L. Heller, Sarah M. Karpanty, Jonathan B. Cohen, Daniel H. Catlin, James D. Fraser.

**Funding acquisition:** Erin L. Heller, Sarah M. Karpanty, Jonathan B. Cohen, James D. Fraser.

**Investigation:** Erin L. Heller, Sarah M. Karpanty, Jonathan B. Cohen, Daniel H. Catlin, Shannon J. Ritter, Barry R. Truitt, James D. Fraser.

**Methodology:** Erin L. Heller, Sarah M. Karpanty, Jonathan B. Cohen, Daniel H. Catlin, Shannon J. Ritter, James D. Fraser.

**Project administration:** Erin L. Heller, Sarah M. Karpanty, Jonathan B. Cohen, Shannon J. Ritter, James D. Fraser.

**Resources:** Erin L. Heller, Sarah M. Karpanty, Jonathan B. Cohen, Barry R. Truitt, James D. Fraser.

**Software:** Erin L. Heller, Jonathan B. Cohen, Daniel H. Catlin, Shannon J. Ritter.

**Supervision:** Erin L. Heller, Sarah M. Karpanty, Jonathan B. Cohen, James D. Fraser.

**Validation:** Erin L. Heller, Sarah M. Karpanty, Jonathan B. Cohen, Daniel H. Catlin, James D. Fraser.

**Visualization:** Erin L. Heller, Sarah M. Karpanty, Jonathan B. Cohen, Daniel H. Catlin, Shannon J. Ritter, James D. Fraser.

**Writing – original draft:** Erin L. Heller.

**Writing – review & editing:** Erin L. Heller, Sarah M. Karpanty, Jonathan B. Cohen, Daniel H. Catlin, Shannon J. Ritter, Barry R. Truitt, James D. Fraser.

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
