## [Decision Letter · Decision Letter 0]

20 Sep 2021

PONE-D-21-26612Factors that affect migratory Western Atlantic Red Knots (Calidris canutus rufa) and their prey during Spring stopover on Virginia’s barrier islandsPLOS ONE

Dear Dr. Karpanty,

Thank you for submitting your manuscript to PLOS ONE. After careful consideration, we feel that it has merit but does not fully meet PLOS ONE’s publication criteria as it currently stands. Therefore, we invite you to submit a revised version of the manuscript that addresses the points raised during the review process.

We look forward to receiving your revised manuscript.

Kind regards,

Vitor Hugo Rodrigues Paiva, Ph.D.

Academic Editor

PLOS ONE

Journal Requirements:

2. We note that Figure 1 in your submission contain map images which may be copyrighted. All PLOS content is published under the Creative Commons Attribution License (CC BY 4.0), which means that the manuscript, images, and Supporting Information files will be freely available online, and any third party is permitted to access, download, copy, distribute, and use these materials in any way, even commercially, with proper attribution. For these reasons, we cannot publish previously copyrighted maps or satellite images created using proprietary data, such as Google software (Google Maps, Street View, and Earth). For more information, see our copyright guidelines: http://journals.plos.org/plosone/s/licenses-and-copyright.

a) You may seek permission from the original copyright holder of Figure 1 to publish the content specifically under the CC BY 4.0 license.  

Reviewers' comments:

Reviewer's Responses to Questions

**Comments to the Author**

1. Is the manuscript technically sound, and do the data support the conclusions?

Reviewer #1: Yes

Reviewer #2: Yes

Reviewer #3: No

2. Has the statistical analysis been performed appropriately and rigorously? 

Reviewer #1: Yes

Reviewer #2: Yes

Reviewer #3: I Don't Know

3. Have the authors made all data underlying the findings in their manuscript fully available?

Reviewer #1: Yes

Reviewer #2: Yes

Reviewer #3: No

4. Is the manuscript presented in an intelligible fashion and written in standard English?

Reviewer #1: Yes

Reviewer #2: Yes

Reviewer #3: Yes

5. Review Comments to the Author

Reviewer #1: The authors used long-term data to explore the factors influencing the red knot abundance and their prey abundance in an important stopover site. This study improves our understanding of habitat selection of red knots, and the influence of prey abundance and composition on red knot presence and flock size. The paper is well-written, the analysis methods are appropriately, and the discussion is comprehensive. I only have several small comments.

Methods: Why did you divide migration period into early and peak migration? Why did you integrate them? Is there previous study suggest that there is great difference between the two periods?

Lines 150-154: You did not clarify why you sampled peat banks more intensive than sand. You mentioned that peat bank is tide-dependent with limited distribution, but I think that’s not enough to illustrate why you mainly focus on peat banks.

Lines 578-583: You conclude that red knots feed on prey that are most digestible, however, I did not see such results. You only mentioned that sites used by red knots contained higher prey abundances than unused sites. You seem did not compare the digestibility of foods used and unused by red knots.

Lines 592-596: So the substrate sampling bias affect the analysis results, as I mentioned above, why you sampled peat bank move intensive than sand.

Lines 705-714: Is the water temperature in your studying sites showed any trend?

Lines 748-756: Is the prey abundance in your studying sites showed any trend?

Reviewer #2: General comments: A commendable study! Invert sampling, especially over such a time series as presented here, is a significant undertaking, but essential to understanding space use of migrants. I thank the authors for the opportunity to review this manuscript. The authors present an impressive time-series dataset and analysis that assess a suite of factors influencing red knot abundance and abundance of their prey at Virginia barrier islands. I have included more detailed comments, by line number below, for consideration by the authors. My overall impression is that this manuscript meets PLOS ONE’s criteria for publication and I agree that it should be published after some minor changes and consideration for shortening and tightening up the manuscript overall.

Line 42: should this more accurately be described as a staging site (Warnock 2010 https://doi.org/10.1111/j.1600-048X.2010.05155.x)? Might be good to clarify this early (not necessarily in this line, maybe lines 55-64 is more appropriate), since the duration of stay and the predictability of food at barrier Islands in Virginia classify this as a staging site (according to Warnock’s argument), rather than a stopover site.

Line 97-99: objective 5 is difficult to follow. I understand the intent, but the sentence structure makes it difficult to wrap my head around. For example, how do you tease apart the prey vs flyway effect? Is that even possible? Consider rephrasing for clarity and consider separating the prey vs. number in the flyway elements.

Line 99: what evidence do you have to support that only the TDF knots use Virginia barrier islands? If birds from northern Brazil or Gulf coast use this space, the presumed flyway population may not be accurate. Consider explaining how this was determined or accounted for in modelling (e.g., lines 245-247). Also, consider adding “boreal” to wintering (or conversely, austral summer), and consider changing throughout. While this is a minor point, this will clarify austral vs boreal winter for our South American friends and not counts of knots that “over-summer” in South America (i.e., they don’t migrate to the Arctic to breed).

Line 163: were flag resightings and flag-to-unflagged ratios collected? If so, these data could be used to better model the passage population using VA’s barrier islands (e.g., MacDonald et al 2020 https://doi.org/10.1002/jwmg.22059), as opposed to extrapolating as described in lines 296-340. In addition, consider clarifying whether all 100 points described in

line 158 fall along the waterline. As stated, it’s not clear that all prey sampling locations also had knots counted, or if counts were only at a select number of points along the waterline.

Line 176: remove “)” after “[…] identify to species.”

Line 177: how do miscellaneous and miscellaneous – horseshoe crab eggs relate? In lines 223-224, the difference is not clear. Are “other organisms” and miscellaneous – horseshoe crab eggs combined? Consider clarifying.

Line 247: see comment above regarding line 99 and TDF estimates as an index for the flyway population. Another consideration is the declines observed in the TDF population that occurred during this study and how those may influence the number of knots in the flyway available to be counted at VA barrier islands.

Line 301: see comment above for line 163. In addition, there is no accounting for turnover at the study site. How was this considered in the modelling? Consider addressing this specifically in the text.

Lines 327-340: While it’s great that assumptions are outlined so clearly (thank you!), these are pretty big assumptions to make, especially the first assumption. If you have the flag resight data, if would be good to test the assumption that the number of birds entering and exiting is equal. While I agree that what we know about knot staging supports this assumption, there is evidence that annual staging varies in duration with ocean temperature and food availability. For example, in Delaware Bay when ocean temperatures are cooler, the crab spawn occurs later and can change the behaviour of staging knots. Birds will not stage as long because there isn’t sufficient food available, so they leave the area entirely to find alternate food sources.

Lines 356-358: consider clarifying whether the analyses above includes all of those described beginning at line 216, or just the ones under the title “Spatial tidal sampling of prey” (line 350).

Line 512: can you explain how the combined substrate figure (fig. 4a) has lower mean abundance values than the split substrate figure (fig. 4b)? I must be missing something, but would expect the combined values to be higher than the split values.

line 514: remove duplicate “radius”

line 516: remove duplicate “on”

lines 635-646: I question the relationship discussed here, since there is no evidence presented that TDF birds are the only population that stages at VA barrier islands, as noted above for lines 99 and 247. In addition, the contrast to previous studies’ results and the suggested alternate rationale that a few large flocks early in the study drove the relationship needs some additional discussion. Consider revising this paragraph.

Lines 647-666: I’m not sure what point this paragraph is trying to get across. It seems relatively straight forward that substrate is an important variable to explain prey type. There’s also discussion around prey abundance as it relates to substrate, which seems straight forward given the small amount of peat available to knots. Consider tightening this paragraph up a bit and combining with the following paragraph?

Line 715-734: for what it’s worth, mean maximum ground counts on southbound migration for Red Knot in James Bay, Ontario were lower in 2012 and 2013, increasing in 2014-2016 (1,600, 1,700, 3,900, 5,900, 6,700, respectively), which aligns with the decline in counts you report here.

Line 722-734: this is a good example of using multiple methods to confirm estimates – nice work.

Line 782-784: how relevant are the peak banks to knots at the study sites? At ~6% of the area, it seems as though this is only marginally important and without the early vs. peak comparison with sand, it’s difficult to see this conclusion through. Consider revising.

Line 786-791: is this statement suggesting that breeding habitat is a limiting factor for knots? I agree that habitat loss or change at sites across a species range is important to understand, and the focus on range-wide studies is great, but it’s surprising that there is no mention, specifically, of the importance of undisturbed, high-quality staging habitat for knots, especially given the study site in VA. Consider revising.

Reviewer #3: I applaud and really appreciate the attempts to come to grips with the potential prey for red knots on the Atlantic coast of Virginia and the attempts to relate measures of abunace of red knots with estimates of their food source, but this was a very very difficult paper for me to review. The problem is that I could not get clear how the work was done. The sampling strategies, both for red knots and their potentiual prey (is there knowledge on their diets here) were poorly described, also in geographic terms. I could simply not picture what had been done, but was then confronted with a lot of statistical detail that almost seemd to ‘replace’ a decent outline of the field methodology. All of this makes it impossible for me to judge the Results or the inferences made, and also makes it impossible to properly compare the feeding ecology of red knots on the Atlantic coast with that of red knots on the other side of the Atlantic (which is well studied, and in view of the references the authors are aware of this literature).

More detailed comments:

Line 31: “Miscellaneous prey (x̄ = 18.85/core sample, SE = 0.88)” is very uninformative for an Abstract as the surface area of the core is not presented. In any case, for useful comparison with most of the relevant literature, rework all these density numbers in number per quare meter please.

Line 42: one of the most impressive and complete shorebird migrant studies with relevant measures during a complete annual cycle is Rakhimberdiev et al. (2018) Food abundance at refuelling sites can mitigate Arctic warming effects on a migratory bird. Nature Communications 9, 4263, which would counts as a pertinent citation here

Line 45: It would appear that the following reference is key here: Piersma, T. & Baker, A.J. (2000). Life history characteristics and the conservation of migratory shorebirds. In L.M. Gosling & W.J. Sutherland (Eds.), Behaviour and conservation (pp. 105-124). Cambridge: Cambridge University Press.

Line 52: it seems odd that the scientific paper that first documented the decline and started all this work is not cited: Baker et al. (2004). Rapid population decline in red knots: fitness consequences of decreased refuelling rates and late arrival in Delaware Bay. Proceedings of the Royal Society of London B 271, 875-882.

Line 70: polyphemus is a species name and should not start with CAP

Line 80: An in-depth review of the relationships between food and shorebird numbers was provided by: Piersma, T. (2012) What is habitat quality? Dissecting a research portfolio on shorebirds. Pp. 383-407 in: Birds and Habitat: Relationships in Changing Landscapes, ed. Robert J. Fuller. Cambridge University Press, Cambridge.

Line 93: ‘predicted number’ is confusing here. Why predicted? Number is enough one would say.

Lines 113-117: The map is not very informative as to the food sampling strategy. Use the space taken by the state of Virgina to show maps where precisely the samples were taken in such a way that future researchers can repeat the exercise. Also with details of locations (geographic coordinates etc., dates of sampling) as supplementary material. It would be good to show the whereabouts of all intertidal peat beds.

Lines 136-162: It remains fully unclear to me how and where and when samples were taken, also making it impossible for future researchers to repeat the exercise. Please be detailed here, with maps etc. (see previous comment).

Lines 164-166: I am now very confused: was the sampling strategy fully determined by the red knot presence? And samples only taken to a depth of 3.5 cm? Knots can probe down to 4 cm, which is the usual cut-off point for harvestability in the many NW European and Asian studies on the food of red knots.

Line 166: please give the surface area of the core and recalculate all density values as number per square meter, to make the values comparable with what almost verybody body else in the literature so far has done.

Line 269: Ah, here is a prey density measure per 275 cm2, please transform to square meters.

Lines 482-483: So this is based on some number per year, but how are Delaware numbers composed (on which basis, nothing that I could find in Methods). I wonder whether this should actually be Discussion.

Line 694: redshank (Tringa tetanus) is not tetanus but totanus

Lines 694-695: “crustaceans contained 4 – 54 Kcals/g ash-free dry mass (AFDM, [121]), while bivalves consumed by red knots in Virginia contained between 0.5 – 31 Kcals/g AFDM (J. Cohen, unpublished data).” This makes me raise eyebrows. First, since a long time the scientific literature uses SI units and in this cae it would be kJoules (kJ) rather than kCalories (not also that kilo should be a small k, rather than K). Secondly, these are odd values, and the range is much too large to be believable as the energy content of the AFDM fraction of marine invertebrates is rather invariable! See the study by Zwarts and Wanink (1993) HOW THE FOOD SUPPLY HARVESTABLE BY WADERS IN THE WADDEN SEA DEPENDS ON THE VARIATION IN ENERGY DENSITY, BODY WEIGHT, BIOMASS, BURYING DEPTH AND BEHAVIOUR OF TIDAL-FLAT INVERTEBRATES. Netherlands Journal of Sea Research 31 (4): 441-476 for an amazing complete review.

Lines 705-717: I fail to see how local temperatures can influence the abundance of stationary prey types that occur somewhere because they settled there as spat. Is this a relevant paragraph?

Line 762: The Conclusions are more like an extended Discussion, so why call them so?

6. PLOS authors have the option to publish the peer review history of their article (what does this mean?). If published, this will include your full peer review and any attached files.

Reviewer #1: No

Reviewer #2: **Yes: **Christian Friis

Reviewer #3: No

---

## [Author Response · Author response to Decision Letter 0]

9 May 2022

Editorial Comments and Responses

Thank you for these reminders. We have checked for compliance with formatting requirements.

2. We note that Figure 1 in your submission contain map images which may be copyrighted. All PLOS content is published under the Creative Commons Attribution License (CC BY 4.0), which means that the manuscript, images, and Supporting Information files will be freely available online, and any third party is permitted to access, download, copy, distribute, and use these materials in any way, even commercially, with proper attribution. For these reasons, we cannot publish previously copyrighted maps or satellite images created using proprietary data, such as Google software (Google Maps, Street View, and Earth). For more information, see our copyright guidelines: http://journals.plos.org/plosone/s/licenses-and-copyright.

Thank you for this helpful guidance (some of which was deleted from this response to save space). We have obtained permission from ESRI to use the basemap for this publication in PLOS ONE and we include their letter in our resubmission materials. We also included ESRI’s required language in the figure caption which differs slightly from the text that you suggest above for other copyrighted figures. We would be happy to modify as needed. The inset satellite map showing an example of the locations of the peat bank sample points was obtained from publicly-available NAIP and the geographic locations of all sample points are provided in our publicly-available database which is cited in the text of the manuscript as Karpanty et al. 2022 [51]. 

We did reformat the literature cited extensively as we added new references at the request of the reviewers. In transferring the literature cited back into the document, all of the extensive renumbering was lost. Therefore, we do attach an extra file that is the original literature cited showing all the track changes in case that is needed.

Reviewer #1 Comments and Responses: The authors used long-term data to explore the factors influencing the red knot abundance and their prey abundance in an important stopover site. This study improves our understanding of habitat selection of red knots, and the influence of prey abundance and composition on red knot presence and flock size. The paper is well-written, the analysis methods are appropriately, and the discussion is comprehensive. I only have several small comments.

Thank you for your positive comments and your constructive review! We have done our best to make the edits as you suggested below, but are happy to make further changes as well. Your review has improved our work! In the acknowledgments, we also added that we thank the AE and all reviewers for their constructive feedback.

Methods: Why did you divide migration period into early and peak migration? Why did you integrate them? Is there previous study suggest that there is great difference between the two periods?

Thank you for the opportunity to clarify our field sampling design, as Reviewer 3 had similar questions. We have extensively revised our description of our field methods, see Lines 148-222 in the track changes version, to explain why we sampled peat bank substrate during the early migration period and both peat and sand substrate during peak migration. We would be happy to make further revisions if this is still unclear.

Lines 150-154: You did not clarify why you sampled peat banks more intensive than sand. You mentioned that peat bank is tide-dependent with limited distribution, but I think that’s not enough to illustrate why you mainly focus on peat banks.

We agree that we needed to do better in explaining our logistical constraints and hope that our revisions, detailed above, have clarified this concern.

Lines 578-583: You conclude that red knots feed on prey that are most digestible, however, I did not see such results. You only mentioned that sites used by red knots contained higher prey abundances than unused sites. You seem did not compare the digestibility of foods used and unused by red knots.

Thank you for catching our poor wording and the opportunity to clarify what we did and did not quantify. You are absolutely correct that we did not quantify digestibility or energetic content of the prey, although we hope to pursue that avenue in future work. We did reword these sentences, now on Lines 652-664, to read as follows. “These relationships suggest that red knots decrease energy expenditure and maximize foraging efficiency by foraging in locations that are most profitable (i.e., those that provide the highest abundance of prey in the shortest period [41, 89]). Prior work in other systems has also consistently demonstrated that red knots likely feed on prey that are most digestible [90-94]. For example, in the Dutch Wadden Sea site, which closely resembles the intertidal habitat of Virginia’s barrier islands, red knots selected juvenile edible cockles (Cerastoderma edule) that had thinner shells and proportionately high flesh content to maximize energy intake rates and reduce processing time (i.e., shell digestion [95-96]). Other studies have also demonstrated that shorebirds such as sanderling (Calidris alba), Eurasian oystercatchers (Haematopus ostralegus), and dunlin (Calidris alpina) select prey that maximize their caloric intake [97-99]. Additional study into the caloric content and digestibility of all prey available to red knots in Virginia would help further clarify the relationship among red knots, prey, and habitat selection.” 

Lines 592-596: So the substrate sampling bias affect the analysis results, as I mentioned above, why you sampled peat bank move intensive than sand.

We hope that our clarification in the methods of why we sampled peat substrate only during early migration period helps better explain this part of the discussion. We also have added some clarification of the limitations created by these logistical constraints in the paragraph, now on lines 668 to 681 as follows: “Most prey abundances (i.e., blue mussels, crustaceans, miscellaneous, and all prey) were highest during early migration period when only peat substrates were sampled due to logistical constraints, while red knot flock size was highest during peak migration period when more samples fell on sand than peat substrate as sampling was done irrespective of tide state. This relationship is likely an artifact of the different sampling approaches used during early migration, when we only sampled random points on peat substrate, and peak migration, when we sampled random points on both peat and sand, as we found prey abundance to be higher in peat than in sand. Therefore, direct comparisons of red knot and prey numbers between early and peak migration periods, without accounting for substrate type, cannot be made under our current sampling design. If logistical constraints could be released so that we could equally sample both peat and sand substrates during both the early and peak migration periods, then more direct comparisons between red knot numbers and prey abundances on the two substrate types could be made.” 

Lines 705-714: Is the water temperature in your studying sites showed any trend?

In another chapter of Heller’s 2021 dissertation, she does report that mean sea surface temperature in April and May showed no linear trend between 2010 – 2018 from data collected at both a buoy at the mouth of the Delaware Bay and Cape Henry, VA. Reviewer 3 suggested that we cut this paragraph, and so we accepted that edit. If you and the editor feel that including this information or keeping this paragraph are important, we would be glad to return this information to the paper.

Lines 748-756: Is the prey abundance in your studying sites showed any trend?

We have found that the abundance of blue mussels in Virginia is declining over the course of our study, and we attribute that to the continued northward retreat of blue mussel breeding areas as detailed in Heller 2021 [42] and another chapter of her dissertation that is in review at another journal. We do not see the same trend for other prey in this system used by red knots. We did include year as a random effect in our modelling efforts to help control for any possible trends over time in the relationship between red knots and prey. We do add this information more clearly into the discussion where we discuss red knots’ use of crustaceans, blue mussels and coquina clams, now on lines 782-787. We would welcome other suggestions on how to incorporate this information if you feel that more is needed.

Reviewer #2 Comments and Responses: General comments: A commendable study! Invert sampling, especially over such a time series as presented here, is a significant undertaking, but essential to understanding space use of migrants. I thank the authors for the opportunity to review this manuscript. The authors present an impressive time-series dataset and analysis that assess a suite of factors influencing red knot abundance and abundance of their prey at Virginia barrier islands. I have included more detailed comments, by line number below, for consideration by the authors. My overall impression is that this manuscript meets PLOS ONE’s criteria for publication and I agree that it should be published after some minor changes and consideration for shortening and tightening up the manuscript overall.

Thank you for your positive comments and your constructive review! We have done our best to make the edits as you suggested below, but are happy to make further changes. Your review has improved our work!

Line 42: should this more accurately be described as a staging site (Warnock 2010 https://doi.org/10.1111/j.1600-048X.2010.05155.x)? Might be good to clarify this early (not necessarily in this line, maybe lines 55-64 is more appropriate), since the duration of stay and the predictability of food at barrier Islands in Virginia classify this as a staging site (according to Warnock’s argument), rather than a stopover site.

We are very grateful for you raising this question and edit. We very much agree that staging site is a better term based on Warnock’s (2010) definition. We did first make this change in the introduction, now line 46, and provide the citation [8] there. We then did a search and find all and replaced ‘stopover’ with ‘staging’ wherever appropriate throughout the manuscript.

Line 97-99: objective 5 is difficult to follow. I understand the intent, but the sentence structure makes it difficult to wrap my head around. For example, how do you tease apart the prey vs flyway effect? Is that even possible? Consider rephrasing for clarity and consider separating the prey vs. number in the flyway elements.

Thank you for this suggestion as we agree that we were testing the prey and the flyway metrics (TDF counts) as independent variables. We did not test for interactions. Thus, we separated this into two objectives, one for prey and one for the TDF, now on Lines 102-104.

Line 99: what evidence do you have to support that only the TDF knots use Virginia barrier islands? If birds from northern Brazil or Gulf coast use this space, the presumed flyway population may not be accurate. Consider explaining how this was determined or accounted for in modelling (e.g., lines 245-247). Also, consider adding “boreal” to wintering (or conversely, austral summer), and consider changing throughout. While this is a minor point, this will clarify austral vs boreal winter for our South American friends and not counts of knots that “over-summer” in South America (i.e., they don’t migrate to the Arctic to breed).

Thank you for all of these helpful suggestions. First, we did not mean to imply that only TDF knots use the Virginia barrier islands in our objectives nor throughout other parts of the paper. In fact, we do know from other formal resighting surveys (see Smith 2008) and our own informal resighting surveys that knots from many different locations are using Virginia barrier islands. For example, Smith 2008 found that knots in Virginia included “the full spectrum of color-flagged knots that one could encounter. Birds were recorded from Argentina, the Canadian Arctic, Brazil, Chile, and the U.S.A.” We have made a number of edits in the manuscript to reflect this helpful edit and to include this Smith 2008 citation. First, on line 102-105 in our objectives, we clarify that we are using TDF knots as an index for the larger flyway population. Then, in the methods, now on lines 303-310 we clarified why we chose to use the TDF counts. This section now reads “The counts of red knots at the Tierra del Fuego boreal wintering grounds served as an index for the total number of red knots in the flyway [21-22]. Red knots marked in locations across the flyway have been resighted on Virginia’s barrier islands [69] but long-term and robust counts of red knots in other locations outside of Delaware Bay were unavailable for comparison across the time span of this study. Thus, the counts of Tierra del Fuego birds were used as an index of red knots in the flyway since the Tierra del Fuego count data fully overlaps the time period of this study and was independent from the U.S. mid-Atlantic Coast region in which both Delaware Bay and Virginia staging sites exist.” We are very open to further edits and appreciate the suggestion to clarify these points. 

Finally, thank you for reminding us to specify that we mean boreal winter and it was silly of us to not clarify this point. We agree that this is an important point to clarify for this species! We make this correction throughout the document as appropriate.

Line 163: were flag resightings and flag-to-unflagged ratios collected? If so, these data could be used to better model the passage population using VA’s barrier islands (e.g., MacDonald et al 2020 https://doi.org/10.1002/jwmg.22059), as opposed to extrapolating as described in lines 296-340. In addition, consider clarifying whether all 100 points described in line 158 fall along the waterline. As stated, it’s not clear that all prey sampling locations also had knots counted, or if counts were only at a select number of points along the waterline.

We very much appreciate this suggestion. We did not systematically conduct flag resightings or collect flag to unflagged ratios. However, we agree with your assessment that this would give a statistically robust estimate of total staging population size, as per the reference that you provided. Indeed, we did pilot a study such as this one year but found that we simply could not do both that study and the one reported here with the resources available. As I noted in the revised methods, the remote locations and challenging boating to these islands requires a crew of two people plus a captain to sample each island per day. We simply have not yet had the funding resources needed to double our crew size in order to accomplish a robust flag-resight study in addition to the data collection described here. However, we do hope eventually to find funding to do this study. We do make some changes in the methods section, now on line 359, to clarify that this is a peak count number, which is the wording used by Lyons 2019 to compare his mark-resight estimates to the aerial and ground peak counts in the Delaware Bay staging site. We also make this change throughout the paper as appropriate so that it is clear that this is not a total staging population estimate.

We did also add clarification, in our field methods revision described above, that all points were along the waterline and that we sampled for red knots and their prey at all points. We hope that this and other revisions suggested by all 3 reviewers make our field methods more understandable and would be happy to make further edits as needed.

Line 176: remove “)” after “[…] identify to species.”

Thank you for catching this error, we have fixed this now.

Line 177: how do miscellaneous and miscellaneous – horseshoe crab eggs relate? In lines 223-224, the difference is not clear. Are “other organisms” and miscellaneous – horseshoe crab eggs combined? Consider clarifying.

Thank you for catching this lack of clarity on our part. Yes, the ‘other organisms’ and miscellaneous are combined into one category. We corrected this, now on lines 228-231, to read: We sorted prey by category (i.e., crustacean (Orders Amphipoda and Calanoida [41]) blue mussel, coquina clam, miscellaneous – horseshoe crab eggs, angel wing clams (Cyrtopleura costata), insect larvae, snails, worms that we were unable to identify to species). We grouped these organisms as “miscellaneous” due to the relatively low number of animals of each type collected.

Line 247: see comment above regarding line 99 and TDF estimates as an index for the flyway population. Another consideration is the declines observed in the TDF population that occurred during this study and how those may influence the number of knots in the flyway available to be counted at VA barrier islands.

We appreciate this point as well. In considering this comment, we looked back at the TDF numbers as reported in USFWS (2019) and note that over the time period of this study (2007-2018) that there were declines in TDF numbers (from 17,360 knots in 2007 to 9,840 knots in 2018), but of course not the dramatic crash seen between the 1990s and early 2000s. Your point is very well-taken, but since this is being used as an index of knot numbers in the flyway, we feel that it is reasonable to assume that the overall flyway population has indeed declined during this time and wanted to ask if this decline did relate the presence and flock size of red knots in Virginia. At this time, we did not make further edits to the manuscript as we did clarify in our earlier responses that we see TDF as an index of flyway populations. We would be happy to consider further revisions.

Line 301: see comment above for line 163. In addition, there is no accounting for turnover at the study site. How was this considered in the modelling? Consider addressing this specifically in the text.

Thank you again for this suggestion to clarify that these are peak count numbers and not total staging population size. We have made edits as described above to clarify that these are peak count numbers here and in methods.

Lines 327-340: While it’s great that assumptions are outlined so clearly (thank you!), these are pretty big assumptions to make, especially the first assumption. If you have the flag resight data, if would be good to test the assumption that the number of birds entering and exiting is equal. While I agree that what we know about knot staging supports this assumption, there is evidence that annual staging varies in duration with ocean temperature and food availability. For example, in Delaware Bay when ocean temperatures are cooler, the crab spawn occurs later and can change the behaviour of staging knots. Birds will not stage as long because there isn’t sufficient food available, so they leave the area entirely to find alternate food sources.

We agree with your points here and we did add a caveat to this assumptions paragraph, now on lines 395-397 to emphasize that these are peak count estimates and not total staging population estimates as in other studies of red knots. This now reads: “The resulting counts of red knots in Virginia during peak migration using both methods make three assumptions and should not be interpreted as total staging population estimates [29, 78] during peak migration until these assumptions are further tested.” We definitely will prioritize finding funding to conduct mark-resight work in future years to run concurrent with these methods.

Lines 356-358: consider clarifying whether the analyses above includes all of those described beginning at line 216, or just the ones under the title “Spatial tidal sampling of prey” (line 350).

Thank you for asking for this clarification. We did clarify, now on line 427, that these R packages were used for all analyses described above and separated this sentence from the preceding paragraph to hopefully further clarify this point.

Line 512: can you explain how the combined substrate figure (fig. 4a) has lower mean abundance values than the split substrate figure (fig. 4b)? I must be missing something, but would expect the combined values to be higher than the split values.

Thank you for this question. This is an outcome of different sample sizes on peat versus sand substrate. On average, there are dramatically higher prey abundances of some types (blue mussels, crustaceans) on the peat substrate but much lower sample sizes on peat substrate than sand substrate. Thus, the inclusion of the peat samples into the combined substrate graph pulls the average down from what it is on its own (Fig 4b) because we had 71 samples on peat banks and 1251 samples on sand during the peak migration period shown in this figure as a result of the sampling design during peak migration being independent of tide. We hope that this clarifies your question. We did check our calculations as well to make sure that we had not made a mistake.

line 514: remove duplicate “radius”

Thank you for catching this error. It has been removed.

line 516: remove duplicate “on”

Thank you for catching this error. It has been removed.

lines 635-646: I question the relationship discussed here, since there is no evidence presented that TDF birds are the only population that stages at VA barrier islands, as noted above for lines 99 and 247. In addition, the contrast to previous studies’ results and the suggested alternate rationale that a few large flocks early in the study drove the relationship needs some additional discussion. Consider revising this paragraph.

Thank you for the opportunity to revise this paragraph. We do hope that we have addressed your concern about our use of TDF counts in earlier comments. Given that we are using TDF counts as an index of the flyway population available to stop in Virginia, but these are not the only birds available to stop in Virginia, we do think that it is reasonable that if there are more birds present in TDF that there would be a higher probability of sites having knots present in Virginia. Given that we do not provide detailed data on the few large red knot flock sizes in the earlier years of the study, and given that was also when TDF numbers were highest, we remove that sentence. Your line of questioning though made us think a better understanding of annual variation in the arrival of knots from other locations in the boreal wintering areas might better explain red knot flock size that TDF counts alone. We made this suggestion now on lines 734-740: Further assessment of the relative contribution of other boreal wintering populations to the Virginia staging site, as Smith [69] began to detail, could help elucidate this perplexing negative relationship between red knot numbers in Tierra del Fuego and flock size in Virginia. It is possible that the total numbers of red knots coming to Virginia from other boreal wintering sites in the flyway varies annually such that counts of red knots in Tierra del Fuego cannot fully explain both the presence and flock size of red knots knots in Virginia. We welcome further suggestions to improve this discussion. 

Lines 647-666: I’m not sure what point this paragraph is trying to get across. It seems relatively straight forward that substrate is an important variable to explain prey type. There’s also discussion around prey abundance as it relates to substrate, which seems straight forward given the small amount of peat available to knots. Consider tightening this paragraph up a bit and combining with the following paragraph?

Thank you for the opportunity and suggestion to streamline our discussion. We agree that we were a bit verbose here and combined these two paragraphs to focus on simply explaining the preferred substrate of the most important prey items and discussing the role of tide in the availability of the different substrate types and prey. These edits are now on lines 742-761.

Line 715-734: for what it’s worth, mean maximum ground counts on southbound migration for Red Knot in James Bay, Ontario were lower in 2012 and 2013, increasing in 2014-2016 (1,600, 1,700, 3,900, 5,900, 6,700, respectively), which aligns with the decline in counts you report here.

Wow, thank you for sharing this information. This is indeed fascinating! If there is a reference or personal citation that you recommend, we would be glad to include this information as well if you think that it improves the manuscript.

Line 722-734: this is a good example of using multiple methods to confirm estimates – nice work.

Thank you for this thought. We were glad to see these relationships held as well!

Line 782-784: how relevant are the peak banks to knots at the study sites? At ~6% of the area, it seems as though this is only marginally important and without the early vs. peak comparison with sand, it’s difficult to see this conclusion through. Consider revising.

Thank you for this opportunity to explain our thinking. While the peat banks make up a proportionally small area of available shoreline, we feel that they are a key component of the foraging dynamics of red knots in this system. As we noted on lines 762-766: For example, blue mussels and crustaceans were captured in the highest densities within two hours of low tide, corresponding to the time of greatest peat substrate exposure, and coquina clams generally were most abundant during the falling and rising tides when clams engage in tidal migrations. Thus, red knots have different prey available in differing abundances dependent on the tidal state, and if peat banks were not present for the blue mussels to settle on, then we argue the availability of food during low tide would be greatly reduced. We did attempt to clarify this point, now on line 880-884, to state: To continue maximizing the availability of red knot prey across the tidal cycle, and in particular the availability of blue mussel prey which requires peat bank substrate to settle in high densities, ongoing management on Virginia’s barrier islands that discourages beach stabilization and nourishment projects and allows the natural processes of overwash and island transgression should continue.

Line 786-791: is this statement suggesting that breeding habitat is a limiting factor for knots? I agree that habitat loss or change at sites across a species range is important to understand, and the focus on range-wide studies is great, but it’s surprising that there is no mention, specifically, of the importance of undisturbed, high-quality staging habitat for knots, especially given the study site in VA. Consider revising.

We appreciate your comment on this final paragraph of our conclusion section. Upon reviewing our revisions, and in light of reviewer 3’s questions about the intent of the conclusion, we have tried to make the conclusion more focused on management or conservation implications. The final paragraph of our discussion relates to the issues you describe here in your comment, and we did revise that paragraph, now on lines 851-866 to better reflect the importance of breeding, wintering and other staging and stopover sites! This paragraph now reads: “The annual fluctuations in the red knot population that migrates through the mid-Atlantic region [35, 133] and the proportion of red knots that use the Virginia staging site may be related to the quality of Delaware Bay and Virginia staging site habitat within a given year. Our long-term findings regarding the relationship between red knots and their prey in Virginia only explain some of the variation in red knot site use and flock size. Because there is no evidence of extreme red knot population fluctuations since their decline in the mid-1990s (this study, [22-23]), and because prey abundance varies over space and time, we speculate that any variation in the number of red knots using the Virginia barrier islands is at least partly due to the abundance and quality of prey at other locations across the annual cycle, including other staging and stopover areas throughout the Western hemisphere. The factors that affect the presence, flock size, and abundances of long-distance migrants using migratory staging sites grounds are not straightforward, as factors across their boreal wintering, breeding, and other staging grounds likely affect birds year-round [16-17]. Thus, additional studies that link potentially relevant variables across each area within the range that red knots use during their life-cycle are warranted to best design successful management practices and develop conservation priorities range-wide [139].”

Reviewer #3 Comments and Responses: I applaud and really appreciate the attempts to come to grips with the potential prey for red knots on the Atlantic coast of Virginia and the attempts to relate measures of abunace of red knots with estimates of their food source, but this was a very very difficult paper for me to review. The problem is that I could not get clear how the work was done. The sampling strategies, both for red knots and their potentiual prey (is there knowledge on their diets here) were poorly described, also in geographic terms. I could simply not picture what had been done, but was then confronted with a lot of statistical detail that almost seemd to ‘replace’ a decent outline of the field methodology. All of this makes it impossible for me to judge the Results or the inferences made, and also makes it impossible to properly compare the feeding ecology of red knots on the Atlantic coast with that of red knots on the other side of the Atlantic (which is well studied, and in view of the references the authors are aware of this literature).

Thank you for your kind comments about the importance of the paper. We apologize that you struggled to understand our methods, especially our field sampling strategies. We have great respect for the extensive work done on red knots globally and will do better to acknowledge that literature and to compare our results. Your detailed review has greatly improved the manuscript.

More detailed comments:

Line 31: “Miscellaneous prey (x̄ = 18.85/core sample, SE = 0.88)” is very uninformative for an Abstract as the surface area of the core is not presented. In any case, for useful comparison with most of the relevant literature, rework all these density numbers in number per quare meter please.

Thank you for this suggestion to convert our red knot and prey data to be reported per square meter. We agree that this will improve the comparability with other studies globally. We have converted these numbers throughout the paper and we have redone all of the Figures and Supplemental Table 1 so that the prey numbers are now shown per meter squared. We also updated the text throughout to replace prey per core with prey per meter squared. 

Line 42: one of the most impressive and complete shorebird migrant studies with relevant measures during a complete annual cycle is Rakhimberdiev et al. (2018) Food abundance at refuelling sites can mitigate Arctic warming effects on a migratory bird. Nature Communications 9, 4263, which would counts as a pertinent citation here

Thank you for this excellent suggestion to include this impactful article. We have included it, now on Lines 43, and in the literature cited [4].

Line 45: It would appear that the following reference is key here: Piersma, T. & Baker, A.J. (2000). Life history characteristics and the conservation of migratory shorebirds. In L.M. Gosling & W.J. Sutherland (Eds.), Behaviour and conservation (pp. 105-124). Cambridge: Cambridge University Press.

Thank you again for this excellent suggestion. We apologize that we did not include some of these key references and requested the books through our library so we could review. We appreciate your recommendations to add these to the paper. We have included it, now on Lines 47, and in the literature cited [10].

Line 52: it seems odd that the scientific paper that first documented the decline and started all this work is not cited: Baker et al. (2004). Rapid population decline in red knots: fitness consequences of decreased refuelling rates and late arrival in Delaware Bay. Proceedings of the Royal Society of London B 271, 875-882.

Thank you for this suggestion. You are correct that this is an important addition and adds to those of Morrison and Watts that were also included originally. We have added this, now on Line 54, and in the literature cited [24].

Line 70: polyphemus is a species name and should not start with CAP

Thank you for catching this typing error. We have fixed this mistake, now on Line 73.

Line 80: An in-depth review of the relationships between food and shorebird numbers was provided by: Piersma, T. (2012) What is habitat quality? Dissecting a research portfolio on shorebirds. Pp. 383-407 in: Birds and Habitat: Relationships in Changing Landscapes, ed. Robert J. Fuller. Cambridge University Press, Cambridge.

Thank you for this helpful suggestion. We requested this book and read the article and agree that this is an important citation. As a side note, I plan to use this chapter in my conservation biology class to further emphasize issues on habitat quality with which students often struggle! We have made this edit on line 87 and in the literature cited [47].

Line 93: ‘predicted number’ is confusing here. Why predicted? Number is enough one would say.

Thank you for this suggestion. In response to other reviewers, we have clarified in the methods that this is an extrapolation/estimation of the peak count of knots using the barrier islands (see reviewer 2) and thus have made this edit to reflect that change, now on line 358. 

Lines 113-117: The map is not very informative as to the food sampling strategy. Use the space taken by the state of Virgina to show maps where precisely the samples were taken in such a way that future researchers can repeat the exercise. Also with details of locations (geographic coordinates etc., dates of sampling) as supplementary material. It would be good to show the whereabouts of all intertidal peat beds.

We appreciate you pushing us to clarify our sampling strategy and again apologize that it was unclear in the initial submission. We have made a number of edits as follows. 

First, the geographic coordinates for every sample point, along with all details of that point including red knot numbers and substrate type and dates, from 2007-2018 are in our publicly available database (Karpanty et al. 2022 [51]). The citation for that database has the DOI to link directly to it. This database includes over 3000 lines of data on the actual sample points and so we think that it is best to reference this database rather than repeat it all in supplemental material. We include this detail in the methods, now on Line 150 and in the Fig. 1 caption. 

Second, we have revised Figure 1 as suggested. We reduced the image of Virginia, as you suggested, and included an inset of a subset of early migration samples on peat banks from 2018 on Metompkin Island to illustrate how points are arrayed. We explain this in the figure 1 caption and also reference a new Supplemental Figure 1 which shows an example sample points for red knots and their invertebrate prey from one year, 2018, as an example.

We hope that these edits make our field sampling design more transparent and are very open to additional suggestions and edits.

Lines 136-162: It remains fully unclear to me how and where and when samples were taken, also making it impossible for future researchers to repeat the exercise. Please be detailed here, with maps etc. (see previous comment).

Please see my edits to the comment above. We are welcome to additional suggested edits as well. We are grateful as your comments and those of the reviewers really pushed us to clarify our sampling methods.

Lines 164-166: I am now very confused: was the sampling strategy fully determined by the red knot presence? And samples only taken to a depth of 3.5 cm? Knots can probe down to 4 cm, which is the usual cut-off point for harvestability in the many NW European and Asian studies on the food of red knots.

We again apologize for misunderstanding and welcome the opportunity to revise this manuscript. Upon re-reading this sentence, we see that it was worded to suggest that we only took prey at points where there were red knots which was not the case. We have reworded this, now on lines 208-219, to clarify that “Field sampling crews would navigate to each random point and count the number of non-flying red knots, if present, within a 100 m radius semicircle of each point placed on the water line. After red knots were counted, if present, or immediately if no red knots were present, we sampled invertebrate prey availability by collecting a core sample of the substrate at the water-line at each sampling point using a section of PVC piping (10 cm diameter x 3.5 cm deep; core volume = 275 cm3 ). Thus, invertebrate prey samples were collected at all points, those with red knots present and those without red knots present. We report prey results as number of organisms/m2 to be comparable to other red knot prey studies and thus assume for purpose of calculation that all prey are on the surface so that the area sampled in each core is 0.00785 m2.”

Second, we went back to the original Tomkovich 1992 paper that assessed geographic variability in Knots based on museum skins from which we decided to have our core depths be 3.5 cm. In Table 1 of Tomkovich (1992), he reports that knots from Eastern North America (n=13) had a range of bill lengths, with the mean for males being 35.6 mm +/- 1.2 (SE) and for females 36.9 mm +/- 1.6 (SE). We used these numbers to select our core depth of 3.5 cm. It was enjoyable to re-read this paper as it discusses the variation in bill length among the different sub-populations of knots. Therefore, I suspect that the 4 cm depth used in European studies is based on additional work of the migrants in that region. Given this review, we did not make any edits in the manuscript to this point but we would be happy to add in these details from the Tomkovich (1992) paper to the methods if the reviewer and editor feel that would improve the manuscript.

Line 166: please give the surface area of the core and recalculate all density values as number per square meter, to make the values comparable with what almost verybody body else in the literature so far has done.

As noted above, we have converted our prey densities to be reported as organisms per meter squared based on this suggestion. We have made edits throughout the manuscript as needed, and in the methods on Lines 219-221, explain that “We report prey results as number of organisms/m2 to be comparable to other red knot prey studies and thus assume for purpose of calculation that all prey are on the surface so that the area sampled in each core is 0.00785 m2.”

Line 269: Ah, here is a prey density measure per 275 cm2, please transform to square meters.

Please see prior comments. We appreciate this suggestion and have made these edits throughout the manuscript.

Lines 482-483: So this is based on some number per year, but how are Delaware numbers composed (on which basis, nothing that I could find in Methods). I wonder whether this should actually be Discussion.

Thank you for catching this omission in our methods. We now explain, on lines 390 to 394, that we chose to use the peak counts from Delaware Bay as those data span the entire range of our data collection, and we provide two references from which we calculated these numbers (USFWS 2019 and Lyons 2019). This now reads, “We used aerial flight or ground-based peak count numbers of red knots in Delaware Bay for these comparisons with Virginia red knot peak count numbers as the Delaware Bay peak count numbers were available for the entire time period analyzed here (2007-2018), whereas more recent mark-resight based estimates of red knot total staging population numbers are only available since 2011 [78-80].” 

Because we now explain how this variable was used in our methods, and we report the lack of a correlation between Virginia and Delaware Bay Counts in the results (Lines 555-557), we would prefer to keep this sentence in the discussion as well. However, we are open to further suggested edits. However, we are open to your and the Editor’s thoughts on all of our revisions.

Line 694: redshank (Tringa tetanus) is not tetanus but tetanus

Thank you for catching this error. Clearly this was a mistake where spellcheck corrected it and we missed it. We have made this correction, now on Line 790.

Lines 694-695: “crustaceans contained 4 – 54 Kcals/g ash-free dry mass (AFDM, [121]), while bivalves consumed by red knots in Virginia contained between 0.5 – 31 Kcals/g AFDM (J. Cohen, unpublished data).” This makes me raise eyebrows. First, since a long time the scientific literature uses SI units and in this cae it would be kJoules (kJ) rather than kCalories (not also that kilo should be a small k, rather than K). Secondly, these are odd values, and the range is much too large to be believable as the energy content of the AFDM fraction of marine invertebrates is rather invariable! See the study by Zwarts and Wanink (1993) HOW THE FOOD SUPPLY HARVESTABLE BY WADERS IN THE WADDEN SEA DEPENDS ON THE VARIATION IN ENERGY DENSITY, BODY WEIGHT, BIOMASS, BURYING DEPTH AND BEHAVIOUR OF TIDAL-FLAT INVERTEBRATES. Netherlands Journal of Sea Research 31 (4): 441-476 for an amazing complete review.

We appreciate that your expert eyes caught this issue. It seems that there was an issue in editing, whereas a decimal point was replaced by a dash. When we went back to the original Goss-Custard 1997 paper, we found that we should have said that the Corophium energetic content estimates were on average 4.35Kcal/g AFDW (interestingly they were reported in Kcal in that paper). Converting this from kCal to kJ (by multiplying kC by 4.184) results in a Corophium energetic content on average of 18.20 kJ/g AFDW according to Goss-Custard (1997), cited in the manuscript as [128]. We very much appreciate you sharing the Zwarts and Wanink [129] paper as frankly we had struggled to find published estimates of energetic content of Mytilus edulis. 

Given this recommendation, we now include these peer-reviewed estimates instead of those from our pilot study that we had in the paper, and which we are not confident given that it was a pilot study and upon reflection of this comment. We then reworked this paragraph to focus on the high energetic value of blue mussels, and concerns about whether they might be replaced by other prey if their abundances in Virginia continue to decline as documented in Heller [42]. This revised paragraph, now on lines 780-801, reads: “Crustaceans were abundant on both peat and sand. Heller [42] found that despite the high abundance of crustaceans across the intertidal landscape of Virginia’s barrier islands, red knots selected crustaceans less than expected given their availability. If ocean temperatures continue to warm [125-126], causing further range contraction and decline in the abundance of blue mussels in Virginia [42], red knots may need to rely more heavily on abundant crustaceans or other prey that we grouped in the miscellaneous category (horseshoe crab eggs, angel wing clams, and other organisms (e.g., insect larvae, snails, worms)) in Virginia. However, coquina clams may become larger and/or more abundant in Virginia due to ocean warming, as they grow faster and mature earlier in warmer water [127]. Previous studies estimating the caloric value of crustaceans (Corophium sp.) consumed by redshank (Tringa tetanus), a shorebird of similar size to red knots, found that crustaceans contained on average 18.20 kJ/g ash-free dry weight (AFDW, [128]). In another study of the energetic content of shorebird prey in the Dutch Wadden Sea, Zwarts and Wanink [129] found that the same species of blue mussels consumed by red knots in Virginia, Mytilus edulis, had the highest energetic content of ten tidal invertebrates measured, with blue mussels having on average about 23.4 kJ/g AFDW. These data demonstrate that crustaceans are less energetically dense than bivalves although still a potentially energetically-rich food resource for red knots; however, red knots’ will likely continue to select bivalves as long as they are available given their high energetic content and the tactile foraging adaptation of red knots that enables them to efficiently capture bivalve prey [130-131].

Lines 705-717: I fail to see how local temperatures can influence the abundance of stationary prey types that occur somewhere because they settled there as spat. Is this a relevant paragraph?

Thank you for this suggestion. Based on these comments and those of earlier reviewers, we have cut this paragraph from the discussion as we think it detracts from the main findings of the paper and was not the focus of our study.

Line 762: The Conclusions are more like an extended Discussion, so why call them so?

We did review the guidelines for PLOS ONE and recognize that a conclusion is optional. We had written it so that it was a summary focused on the conservation implications of our key findings. We would be happy to rewrite to make sure that all key points are embedded simply in the discussion, but if the editor agrees, we would prefer to keep it as a stand-alone conclusion.

Editor Comment

Thank you for sharing this tool. On March 15th, 2022 I used this tool with all of our Figures for this resubmission and it did lead me to fix a few issues with dimensions and resolution. All the submitted figures, including the supplemental figure 1, got the green checkmark in this program.

---

## [Decision Letter · Decision Letter 1]

7 Jun 2022

Factors that affect migratory Western Atlantic Red Knots (*Calidris canutus rufa*) and their prey during Spring staging on Virginia’s barrier islands

PONE-D-21-26612R1

Dear Dr. Karpanty,

We’re pleased to inform you that your manuscript has been judged scientifically suitable for publication and will be formally accepted for publication once it meets all outstanding technical requirements.

Kind regards,

Vitor Hugo Rodrigues Paiva, Ph.D.

Academic Editor

PLOS ONE

Additional Editor Comments (optional):

Reviewers' comments:

Reviewer's Responses to Questions

**Comments to the Author**

1. If the authors have adequately addressed your comments raised in a previous round of review and you feel that this manuscript is now acceptable for publication, you may indicate that here to bypass the “Comments to the Author” section, enter your conflict of interest statement in the “Confidential to Editor” section, and submit your "Accept" recommendation.

Reviewer #1: All comments have been addressed

Reviewer #2: All comments have been addressed

2. Is the manuscript technically sound, and do the data support the conclusions?

Reviewer #1: Yes

Reviewer #2: Yes

3. Has the statistical analysis been performed appropriately and rigorously? 

Reviewer #1: Yes

Reviewer #2: Yes

4. Have the authors made all data underlying the findings in their manuscript fully available?

Reviewer #1: Yes

Reviewer #2: Yes

5. Is the manuscript presented in an intelligible fashion and written in standard English?

Reviewer #1: Yes

Reviewer #2: Yes

6. Review Comments to the Author

Reviewer #1: (No Response)

Reviewer #2: Thank you to the authors for addressing my previous comments in the original submission. The revised version has provided the clarity requested along with tightening up the discussion.

7. PLOS authors have the option to publish the peer review history of their article (what does this mean?). If published, this will include your full peer review and any attached files.

Reviewer #1: No

Reviewer #2: **Yes: **Christian Friis

---

## [Editor Report · Acceptance letter]

13 Jun 2022

PONE-D-21-26612R1 

Factors that affect migratory Western Atlantic Red Knots (*Calidris canutus rufa*) and their prey during Spring staging on Virginia’s barrier islands 

Dear Dr. Karpanty:

I'm pleased to inform you that your manuscript has been deemed suitable for publication in PLOS ONE. Congratulations! Your manuscript is now with our production department. 

Kind regards, 

on behalf of

Dr. Vitor Hugo Rodrigues Paiva 

Academic Editor

PLOS ONE